# GATSBI: Generative Adversarial Training for Simulation-Based Inference

**Poornima Ramesh**
University of Tübingen

**Jan-Matthis Lueckmann**
University of Tübingen

**Jan Boelts**
TU Munich

**Álvaro Tejero-Cantero**
University of Tübingen

**David S. Greenberg**
Helmholtz Centre Hereon

**Pedro J. Gonçalves**
University of Tübingen

**Jakob H. Macke**
University of Tübingen

## Abstract

Simulation-based inference (SBI) refers to statistical inference on stochastic models for which we can generate samples, but not compute likelihoods. Like SBI algorithms, generative adversarial networks (GANs) do not require explicit likelihoods. We study the relationship between SBI and GANs, and introduce GATSBI, an adversarial approach to SBI. GATSBI reformulates the variational objective in an adversarial setting to learn implicit posterior distributions. Inference with GATSBI is amortised across observations, works in high-dimensional posterior spaces and supports implicit priors. We evaluate GATSBI on two SBI benchmark problems and on two high-dimensional simulators. On a model for wave propagation on the surface of a shallow water body, we show that GATSBI can return well-calibrated posterior estimates even in high dimensions. On a model of camera optics, it infers a high-dimensional posterior given an implicit prior, and performs better than a state-of-the-art SBI approach. We also show how GATSBI can be extended to perform sequential posterior estimation to focus on individual observations. Overall, GATSBI opens up opportunities for leveraging advances in GANs to perform Bayesian inference on high-dimensional simulation-based models.

## 1 Introduction

Hypothesis-making in many scientific disciplines relies on stochastic simulators that—unlike expressive statistical models such as neural networks—have domain-relevant, interpretable parameters. Finding the parameters $\theta$ of a simulator that reproduce the observed data $x_{\mathrm{o}}$ constitutes an inverse problem. In order to use these simulators to formulate further hypotheses and experiments, one needs to obtain uncertainty estimates for the parameters and allow for multi-valuedness, i.e., different candidate parameters accounting for the same observation. These requirements are met by Bayesian inference, which attempts to approximate the posterior distribution $p(\theta|x_{\mathrm{o}})$. While a variety of techniques exist to calculate posteriors for scientific simulators which allow explicit likelihood calculations $p(x|\theta)$, inference on black-box simulators with intractable likelihoods a.k.a. 'simulation-based inference' (Cranmer et al., 2020), poses substantial challenges. In traditional, so-called Approximate Bayesian Computation (ABC) approaches to SBI (Beaumont et al., 2002, 2009; Marjoram et al., 2003; Sisson et al., 2007), one samples parameters $\theta$ from a prior $\pi(\theta)$ and accepts only those parameters for which the simulation output $x \sim p(x|\theta)$ is close to the observation $d(x, x_{\mathrm{o}}) < \epsilon$. With increasing data dimensionality $N$, this approach incurs an exponentially growing simulation expense ('curse of dimensionality'); it also requires suitable choices of distance function $d$ and acceptance threshold $\epsilon$.

Recent SBI algorithms (e.g., Greenberg et al., 2019; Gutmann and Corander, 2015; Hermans et al., 2020; Lueckmann et al., 2017; Meeds and Welling, 2014; Papamakarios and Murray, 2016; Radev et al., 2020; Thomas et al., 2021) draw on advances in machine learning and are often based on Gaussian Processes (Rasmussen and Williams, 2006) or neural networks for density estimation (e.g. using normalizing flows; Papamakarios et al., 2021). While this has led to substantial improvements over classical approaches, and numerous applications in fields such as cosmology (Cole et al., 2021), neuroscience (Gonçalves et al., 2020) or robotics (Muratore et al., 2021), statistical inference for

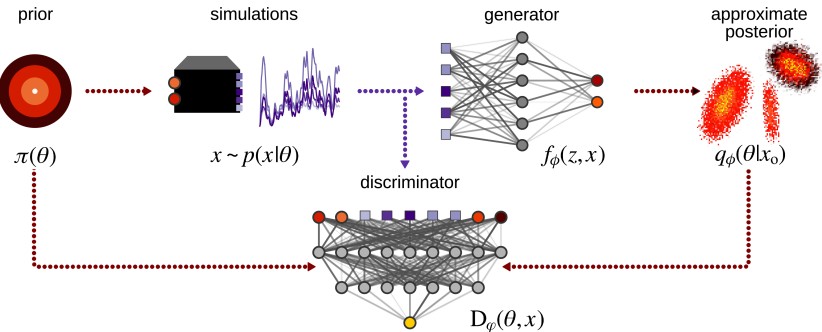

Figure 1: **GATSBI uses a conditional generative adversarial network (GAN) for simulation-based inference (SBI)**. We sample parameters $\theta$ from a prior $\pi(\theta)$, and use them to generate synthetic data $x$ from a black-box simulator. The GAN generator learns an implicit approximate posterior $q_\phi(\theta|x)$, i.e., it learns to generate posterior samples $\theta'$ given data $x$. The discriminator is trained to differentiate between $\theta'$ and $\theta$, conditioned on $x$.

high-dimensional parameter spaces is an open challenge (Cranmer et al., 2020; Lueckmann et al., 2021): This prevents the application of these methods to many real-world scientific simulators.

In machine learning, generative adversarial networks (GANs, Goodfellow et al., 2014) have emerged as powerful tools for learning generative models of high-dimensional data, for instance, images (Isola et al., 2016; Radford et al., 2015; Zhu et al., 2017). Using a minimax game between a 'generator' and a 'discriminator', GANs can generate samples which look uncannily similar to the data they have been trained on. Like SBI algorithms, adversarial training is 'likelihood-free', i.e., it does not require explicit evaluation of a density function, and also relies on comparing empirical and simulated data. This raises the question of whether GANs and SBI solve the same problem, and in particular whether and how adversarial training can help scale SBI to high-dimensional regimes.

Here, we address these questions and make the following contributions: First, we show how one can use adversarial training to learn an implicit posterior density for SBI. We refer to this approach as GATSBI (Generative Adversarial Training for SBI)[1]. Second, we show that GATSBI can be formulated as an adversarial variational inference algorithm (Huszár, 2017; Makhzani et al., 2015; Mescheder et al., 2017), thus opening up a potential area of exploration for SBI approaches. Third, we show how it can be extended to refine posterior estimates for specific observations, i.e., sequential posterior estimation. Fourth, on two low-dimensional SBI benchmark problems, we show that GATSBI's performance is comparable to common SBI algorithms. Fifth, we illustrate GATSBI on high-dimensional inference problems which are out of reach for most SBI methods: In a fluid dynamics problem from earth science, we learn an implicit posterior over 100 parameters. Furthermore, we show that GATSBI (in contrast to most SBI algorithms) can be used on problems in which the *prior* is only specified implicitly: by recasting image-denoising as an SBI problem, we learn a posterior over (784-dimensional) images. Finally, we discuss the opportunities and limitations of GATSBI relative to current SBI algorithms, and chart out avenues for future work.

## 2 METHODS

### 2.1 SIMULATION-BASED INFERENCE AND ADVERSARIAL NETWORKS

**Simulation-based inference** (SBI) targets stochastic simulators with parameters $\theta$ that we can use to generate samples $x$. In many scientific applications, simulators are 'black-box': we cannot evaluate the likelihood $p(x|\theta)$, take derivatives through the simulator, or access its internal random numbers. The goal of SBI is to infer the parameters $\theta$, given empirically observed data $x_o$ i.e., to obtain the posterior distribution $p(\theta|x_o)$. In most applications of existing SBI algorithms, the simulator is a mechanistic forward model depending on a low ($\sim 10$) number of parameters $\theta$.

---

[1]Note that the name is shared by recently proposed, and entirely unrelated, algorithms (Min et al., 2021; Yu et al., 2020)

**Generative adversarial networks (GANs)** (Goodfellow et al., 2014) consist of two networks trained adversarially. The generator $f_\phi(z)$ produces samples $x$ from latent variables $z$; the discriminator $D_\psi$ acts as a similarity measure between the distribution of generated $x$ and observations $x_o$ from a distribution $p_{\text{data}}(x_o)$. The two networks are pitted against each other: the generator attempts to produce samples that would fool the discriminator; the discriminator is trained to tell apart generated and ground-truth samples. This takes the form of a minimax game $\min_\phi \max_\psi L(\phi, \psi)$ between the networks, with cross-entropy as the value function:

$$L(\phi, \psi) = \mathbb{E}_{p_{\text{data}}(x_o)} \log D_\psi(x_o) + \mathbb{E}_{p(z)} \log(1 - D_\psi(f_\phi(z))).$$

At the end of training, if the networks are sufficiently expressive, samples from $f_\phi$ are hard to distinguish from the ground-truth samples, and the generator approximates the observed data distribution $p_{\text{data}}(x_o)$. GANs can also be used to perform (implicit) *conditional* density estimation, by making the generator and discriminator conditional on external covariates $t$ (Mirza and Osindero, 2014). Note that the GAN formulation does not evaluate the intractable distribution $p_{\text{data}}(x_o)$ or $p_{\text{data}}(x_o|t)$ at any point. Thus, GANs and SBI appear to be solving similar problems: They are both 'likelihood-free' i.e., they do not evaluate the target distribution, but rather exclusively work with samples from it.

**Nevertheless, GANs and SBI differ in multiple ways:** GANs use neural networks, and thus are always differentiable, whereas the numerical simulators in SBI may not be. Furthermore, the latent variables for the GAN generator are typically accessible, while in SBI, one may not have access to the simulator's internal random variables. Finally, in GANs, the goal is to match the distribution of the generated samples $x$ to that of the observed data $x_o$, and parameters of the generator are learnt as point estimates. In contrast, the goal of SBI is to learn a distribution over parameters $\theta$ of a simulator consistent with both the observed data $x_o$ *and* prior knowledge about the parameters. However, conditional GANs *can* be used to perform conditional density estimation. How can we repurpose them for SBI? We propose an approach using adversarial training for SBI, by employing GANs as implicit density estimators rather than generative models of data.

## 2.2 Generative Adversarial Training for SBI: GATSBI

Our approach leverages conditional GANs for SBI. We train a deep generative neural network $f_\phi$ with parameters $\phi$, i.e., $f$ takes as inputs observations $x$ and noise $z$ from a fixed source with distribution $p(z)$, so that $f_\phi(x, z)$ deterministically transforms $z$ to generate $\theta$. This generative network $f$ is an implicit approximation $q_\phi(\theta|x)$ of the posterior distribution, i.e.,

$$\left. \begin{array}{l} z \sim p(z) \\ \theta = f_\phi(z, x) \end{array} \right\} \Rightarrow \theta \sim q_\phi(\theta|x). \tag{1}$$

We train $f$ adversarially: We sample many parameters $\theta$ from the prior, and for each $\theta$, simulate $x \sim p(x|\theta)$ i.e., we sample tuples $(\theta, x) \sim \pi(\theta)p(x|\theta) = p(\theta|x)p(x)$. These samples from the joint constitute our training data. We define a discriminator D, parameterised by $\psi$, to differentiate between samples $\theta$ drawn from the approximate posterior $q_\phi(\theta|x)$ and the true posterior $p(\theta|x)$. The discriminator is conditioned on $x$. We calculate the cross-entropy of the discriminator outputs, and maximize it with respect to the discriminator parameters $\psi$ and minimize it with respect to the generator parameters $\phi$ i.e., $\min_\phi \max_\psi L(\phi, \psi)$, with

$$L(\phi, \psi) = \mathbb{E}_{\pi(\theta)p(x|\theta)p(z)} \Big[ \log D_\psi(\theta, x) + \log \big(1 - D_\psi(f_\phi(z, x), x)\big) \Big]$$

$$= \mathbb{E}_{p(x)} \Big[ \mathbb{E}_{p(\theta|x)} \big( \log D_\psi(\theta, x) \big) + \mathbb{E}_{q(\theta|x)} \big( \log \big(1 - D_\psi(\theta, x)\big) \big) \Big] \tag{2}$$

This conditional GAN value function targets the desired posterior $p(\theta|x)$:

**Proposition 1.** *Given a fixed generator $f_\phi$, the discriminator $D_{\psi^*}$ maximizing equation 2 satisfies*

$$D_{\psi^*}(\theta, x) = \frac{p(\theta|x)}{p(\theta|x) + q_\phi(\theta|x)}, \tag{3}$$

*and the corresponding loss function for the generator parameters is the Jensen-Shannon divergence (JSD) between the true and approximate posterior,*

$$L_{\psi^*}(\phi) = 2 \operatorname{JSD}(p(\theta|x)||q_\phi(\theta|x)) - \log 4.$$

Using sufficiently flexible networks, the GATSBI generator trained with the cross-entropy loss converges in the limit of infinitely many samples to the true posterior, since the JSD is minimised if and only if $q_\phi(\theta|x) = p(\theta|x)$. The proof directly follows Prop. 1 and 2 from Goodfellow et al. (2014) and is given in detail in App. A.1. GATSBI thus performs *amortised inference*, i.e., learns a single inference network which can be used to perform inference rapidly and efficiently for a wide range of potential observations, without having to re-train the network for each new observation. The training steps for GATSBI are described in App. B.

## 2.3 RELATING GATSBI TO EXISTING SBI APPROACHES

There have been previous attempts at using GANs to fit black-box scientific simulators to empirical data. However, unlike GATSBI, these approaches do not perform Bayesian inference: Kim et al. (2020) and Louppe et al. (2017) train a discriminator to differentiate between samples from the black-box simulator and empirical observations. They use the discriminator to adjust a proposal distribution for simulator parameters until the resulting simulated data and empirical data match. However, neither approach returns a posterior distribution representing the balance between a prior distribution and a fit to empirical data. Similarly, Jethava and Dubhashi (2017) use two generator networks, one to approximate the prior and one to learn summary statistics, as well as two discriminators. This scheme yields a generator for parameters leading to realistic simulations, but does not return a posterior over parameters. Parikh et al. (2020) train a conditional GAN (c-GAN), as well as two additional GANs (r-GAN and t-GAN) within an active learning scheme, in order to solve 'population of models' problems (Britton et al., 2013; Lawson et al., 2018). While c-GAN exhibits similarities with GATSBI, the method does not do Bayesian inference, but rather solves a constrained optimization problem. Adler and Öktem (2018) use Wasserstein-GANs (Arjovsky et al., 2017) to solve Bayesian inverse problems in medical imaging. While their set-up is similar to GATSBI, the Wasserstein metric imposes stronger conditions for the generator to converge to the true posterior (App. Sec. A.3).

Several SBI approaches train discriminators to circumvent evaluation of the intractable likelihood (e.g., Cranmer et al., 2015; Gutmann et al., 2018; Pham et al., 2014; Thomas et al., 2021). Hermans et al. (2020) train density-ratio estimators by discriminating samples $(\theta, x) \sim \pi(\theta)p(x|\theta)$ from $(\theta, x) \sim \pi(\theta)p(x)$, which can be used to sample from the posterior. Unlike GATSBI, this requires a potentially expensive step of MCMC sampling. A closely related approach (Durkan et al., 2020; Greenberg et al., 2019) directly targets the posterior, but requires a density estimator that can be evaluated—unlike GATSBI, where the only restriction on the generator is that it remains differentiable.

Finally, ABC and synthetic likelihood methods have been developed to tackle problems in high-dimensional parameter spaces (>100-dimensions) although these require model assumptions regarding the Gaussianity of the likelihood (Ong et al., 2018), or low-dimensional summary statistics (Kousathanas et al., 2015; Rodrigues et al., 2019), in addition to MCMC sampling.

## 2.4 RELATION TO ADVERSARIAL INFERENCE

GATSBI reveals a direct connection between SBI and adversarial variational inference. Density estimation approaches for SBI usually use the forward Kullback-Leibler divergence ($D_{KL}$). The reverse $D_{KL}$ can offer complementary advantages (e.g. by promoting mode-seeking rather than mode-covering (Turner and Sahani, 2011)), but is hard to compute and optimise for conventional density estimation approaches. We here show how GATSBI, by using an approach similar to adversarial variational inference, performs an optimization that also finds a minimum of the reverse $D_{KL}$.

First, we note that density estimation approaches for SBI involve maximising the approximate log posterior over samples simulated from the prior. This is equivalent to minimising the *forward* $D_{KL}$:

$$L_{\text{fwd}}(\phi) = \mathbb{E}_{p(x)} D_{\text{KL}}(p(\theta|x)||q_\phi(\theta|x)) = -\mathbb{E}_{p(x)p(\theta|x)} \log q_\phi(\theta|x) + \mathbb{E}_{p(x)p(\theta|x)} \log p(\theta|x). \quad (4)$$

rather than reverse $D_{KL}$:

$$L_{\text{rev}}(\phi) = \mathbb{E}_{p(x)} D_{\text{KL}}(q_\phi(\theta|x)||p(\theta|x)) = \mathbb{E}_{p(x)q_\phi(\theta|x)} \log \frac{q_\phi(\theta|x)}{p(\theta|x)}, \quad (5)$$

This is because it is feasible to compute $L_{\text{fwd}}$ with an intractable likelihood: it only involves evaluating the approximate posterior under *samples* from the true joint distribution $p(\theta, x) = p(x|\theta)\pi(\theta)$. The problem with the reverse $D_{KL}$, however, is that it requires evaluating the true posterior $p(\theta|x)$ under

samples from the approximate posterior $q_\phi(\theta|x)$, which is impossible with an intractable likelihood. This is a problem also shared by variational-autoencoders (VAEs, Kingma and Welling, 2013) but which can be solved using adversarial inference.

A VAE typically consists of an encoder network $q_\phi(u|x)$ approximating the posterior over latents $u$ given data $x$, and a decoder network approximating the likelihood $p_\alpha(x|u)$. The parameters $\phi$ and $\alpha$ of the two networks are optimised by maximising the Evidence Lower Bound (ELBO):

$$
\begin{aligned}
\text{ELBO}(\alpha, \phi) &= -\mathbb{E}_{p(x)}\big[D_{\text{KL}}(q_\phi(u|x)||p(u)) + \mathbb{E}_{q_\phi(u|x)}[\log p_\alpha(x|u)]\big] \\
&= -\mathbb{E}_{p(x)q_\phi(u|x)} \log \frac{q_\phi(u|x)p(x)}{p(u)p_\alpha(x|u)} + \text{const.} \\
&= -D_{\text{KL}}(q_\phi(u|x)p(x)||p(u)p_\alpha(x|u)) + \text{const.} \quad (6)
\end{aligned}
$$

Note that the $D_{\text{KL}}$ is minimised when maximising the ELBO. If any of the distributions used to compute the ratio in the $D_{\text{KL}}$ in equation 6 is intractable, one can do adversarial inference, i.e., the ratio can be approximated using a discriminator (see Prop. 1) and $q_\phi(u|x)$ is recast as the generator. Various adversarial inference approaches approximate different ratios, depending on which of the distributions is intractable (see App. Table 1 for a non-exhaustive comparison of different discriminators and corresponding losses), and their connection to GANs has been explored extensively (Chen et al., 2018; Hu et al., 2017; Mohamed and Lakshminarayanan, 2016; Srivastava et al., 2017).

Recasting SBI into the VAE framework, the intractable distribution is the likelihood $p_\alpha(x|\theta)$,[2] with $\alpha$ constant, i.e., the simulator only has parameters to do inference over. Hence, $p(x)$ is defined as the marginal likelihood. In this setting, the SBI objective becomes the reverse $D_{\text{KL}}$, and we can use an adversarial method, e.g. GATSBI, to minimise it for learning $q_\phi(\theta|x)$. There is an additional subtlety at play here: GATSBI's generator loss, given an optimal discriminator, is not the reverse $D_{\text{KL}}$, but the JSD (see Prop. 1). However, following the proof for Prop. 3 in Mescheder et al. (2017), we show that the approximate posterior that minimises the JSD *also* maximises the ELBO (see App. A.2). Thus, GATSBI is an adversarial inference method that performs SBI using the *reverse* $D_{\text{KL}}$.

Note that the converse is not true: not all adversarial inference methods can be adapted for SBI. Specifically, some adversarial inference methods require explicit evaluation of the likelihood. Of those methods listed in Table 1, only likelihood-free variational inference (LFVI) (Tran et al., 2017) is explicitly used to perform inference for a simulator with intractable likelihood. LFVI is defined as performing inference over latents $u$ and global parameters $\beta$ shared across multiple observations $x$, which are sampled from an empirical distribution $p'(x) \neq p(x)$, i.e., it learns $q_\phi(u, \beta|x)$. However, if $p'(x) = p(x)$, and $\beta$ is constant, i.e., the simulator does not have tunable parameters (but only parameters one does inference over) then LFVI and GATSBI become (almost) equivalent: LFVI differs by a single term in the loss function, and we empirically found the methods to yield similar results (see App. Fig. 6). We discuss LFVI's relationship to GATSBI in more detail in App. A.3.

## 2.5 SEQUENTIAL GATSBI

The GATSBI formulation in the previous sections allows us to learn an amortised posterior $p(\theta|x)$ for *any* observation $x$. However, one is often interested in good posterior samples for a particular experimental observation $x_{\text{o}}$—and not for all possible observations $x$. This can be achieved by training the density estimator using samples $\theta$ from a proposal prior $\tilde{\pi}(\theta)$ instead of the usual prior $\pi(\theta)$. The proposal prior ideally produces samples $\theta$ that are localised around the modes of $p(\theta|x_{\text{o}})$ and can guide the density estimator towards inferring a posterior for $x_{\text{o}}$ using less training data. To this end, sequential schemes using proposal distributions to generate training data over several rounds of inference (e.g. by using the approximate posterior from previous rounds as the proposal prior for subsequent rounds) can be more sample efficient (Lueckmann et al., 2021).

Sequential GATSBI uses a scheme similar to other sequential SBI methods (e.g., Greenberg et al., 2019; Hermans et al., 2020; Lueckmann et al., 2017; Papamakarios and Murray, 2016; Papamakarios et al., 2019): the generator from the current round of posterior approximation becomes the proposal distribution in the next round. However, using samples from a proposal prior $\tilde{\pi}(\theta)$ to train the GAN would lead to the GATSBI generator learning a *proposal posterior* $\tilde{p}(\theta|x)$ rather than the true posterior

---

[2]Note the difference in notation: the parameters $\theta$ correspond to the latents $u$ of a VAE, and the simulator in SBI to the decoder of the VAE.

$p(\theta|x)$. Previous sequential methods solve this problem either by post-hoc correcting the learned density (Papamakarios and Murray, 2016), by adjusting the loss function (Lueckmann et al., 2017), or by reparametrizing the posterior network (Greenberg et al., 2019). While these approaches might work for GATSBI (see App. A.4 for an outline), we propose a different approach: at training time, we sample the latents $z$—that the generator transforms into parameters $\theta$—from a *corrected* distribution $p_t(z)$, so that the samples $\theta$ are implicitly generated from an approximate proposal posterior $\tilde{q}_\phi(\theta|x)$:

$$\left.\begin{array}{c} z \sim p_t(z) \\ \theta = f_\phi(z, x) \end{array}\right\} \Rightarrow \theta \sim \tilde{q}_\phi(\theta|x), \tag{7}$$

where $p_t(z) = p(z)\,(\omega(f_\phi(z,x),x))^{-1}$, $\omega(\theta,x) = \frac{\pi(\theta)}{\tilde{\pi}(\theta)}\frac{\tilde{p}(x)}{p(x)}$, and $\tilde{p}(x)$ is the marginal likelihood under samples from the proposal prior. Since $\tilde{\pi}(\theta)$, $p(x)$ and $\tilde{p}(x)$ are intractable, we approximate $\omega(\theta,x)$ using ratio density estimators for $\frac{\pi(\theta)}{\tilde{\pi}(\theta)}$ and $\frac{\tilde{p}(x)}{p(x)}$. This ensures that at inference time, the GATSBI generator transforms samples from $p(z)$ into an approximation of the true posterior $p(\theta|x)$ with highest accuracy at $x = x_o$ (see App. A.4, B). Azadi et al. (2019); Che et al. (2020) use similar approaches to improve GAN training. Note that this scheme may be used with other inference algorithms evaluating a loss under approximate posterior samples, e.g. LFVI (see Sec. 2.4).

## 3 RESULTS

We first investigate GATSBI on two common benchmark problems in SBI (Lueckmann et al., 2021). We then show how GATSBI infers posteriors in a high-dimensional regime where most state-of-the-art SBI algorithms are inadequate, as well as in a problem with an implicit prior (details in App. D.1).

### 3.1 BENCHMARK PROBLEMS

We selected two benchmark problems with low-dimensional posteriors on which state-of-the-art SBI algorithms have been tested extensively. We used the benchmarking-tools and setup from Lueckmann et al. (2021). Briefly, we compared GATSBI against five SBI algorithms: classical rejection ABC (Tavaré et al., 1997, REJ-ABC), sequential Monte Carlo ABC (Beaumont et al., 2009, SMC-ABC) and the single-round variants of flow-based neural likelihood estimation (Papamakarios et al., 2019, NLE), flow-based neural posterior estimation (Greenberg et al., 2019; Papamakarios and Murray, 2016, NPE) and neural ratio estimation (Durkan et al., 2020; Hermans et al., 2020, NRE). GANs have previously been shown to be poor density estimators for low-dimensional distributions (Zaheer et al., 2017). Thus, while we include these examples to provide empirical support that GATSBI works, we do not expect it to perform as well as state-of-the-art SBI algorithms on these tasks. We compared samples from the approximate posterior against a reference posterior using a classifier trained on the two sets of samples (classification-based two-sample test, C2ST). A C2ST accuracy of 0.5 (chance level) corresponds to perfect posterior estimation.

**"Simple Likelihood Complex Posterior" (SLCP)** is a challenging SBI problem designed to have a simple likelihood and a complex posterior (Papamakarios et al., 2019). The prior $\pi(\theta)$ is a 5-dimensional uniform distribution and the likelihood for the 8-dimensional $x$ is a Gaussian whose mean and variance are nonlinear functions of $\theta$. This induces a posterior over $\theta$ with four symmetrical modes and vertical cut-offs (see App. Fig. 7), making it a challenging inference problem. We trained all algorithms on a budget of 1k, 10k and 100k simulations. The GATSBI C2ST scores were on par with NRE, better than REJ-ABC and SMC-ABC, but below the NPE and NLE scores (Figure 2A).

The **"Two-Moons Model"** (Greenberg et al., 2019) has a simple likelihood and a bimodal posterior, where each mode is crescent-shaped (see App. Fig. 8). Both $x$ and $\theta$ are two-dimensional. We initially trained GATSBI with the same architecture and settings as for the SLCP problem. While the classification score for GATSBI (see Figure 2B) was on par with that of REJ-ABC and SMC-ABC for 1k simulations, performance did not significantly improve when increasing training budget. When we tuned the GAN architecture and training hyperparameters specifically for this model, we found a performance improvement. Lueckmann et al. (2021) showed that performance can be significantly improved with sequential inference schemes. We found that although using sequential schemes with GATSBI leads to small performance improvements, it still did not ensure performance on-par with the flow-based methods (see App. Fig. 9). Moreover, since it had double the number of hyperparameters as amortised GATSBI, sequential GATSBI also required more hyperparameter tuning.

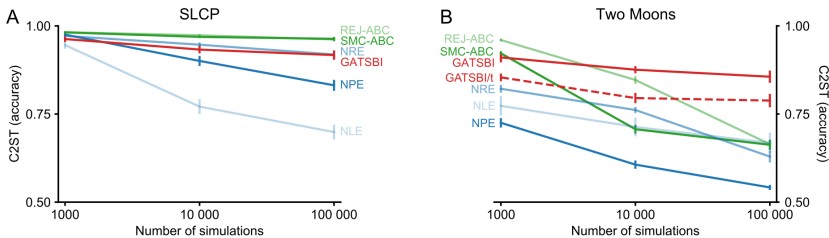

Figure 2: **GATSBI performance on benchmark problems.** Mean C2ST score (± standard error of the mean) for 10 observations each. A. The classification score for GATSBI (red) on SCLP decreases with increasing simulation budget, and is comparable to NRE. It outperforms rejection ABC and SMC-ABC, but has worse performance than NPE and NLE. B. The classification score for GATSBI (red) on the two-moons model decreases with increasing simulation budget, is comparable to REJ-ABC and SMC-ABC for simulation budgets of 1k and 10k, and is outperformed by all other methods for a 100k simulation budget. However, GATSBI's classification score improves when its architecture and optimization parameters are tuned to the two-moons model (red, dashed).

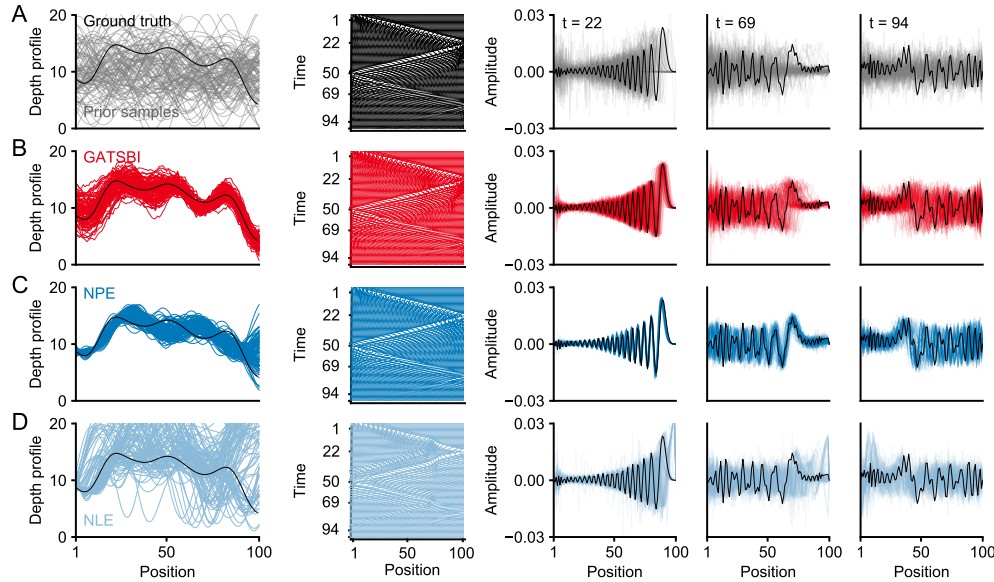

Figure 3: **Shallow water model inference with GATSBI, NPE and NLE.** A. Ground truth, observation and prior samples. Left: ground-truth depth profile and prior samples. Middle: surface wave simulated from ground-truth profile as a function of position and time. Right: wave amplitudes at three different fixed times for ground-truth depth profile (black), and waves simulated from multiple prior samples (gray). B. GATSBI inference. Left: posterior samples (red) versus ground-truth depth profile (black). Middle: surface wave simulated from a single GATSBI posterior sample. Right: wave amplitudes for multiple GATSBI posterior samples, at three different times (red). Ground-truth waves in black. C. NPE inference. Panels as in B. D. NLE inference.

## 3.2 HIGH-DIMENSIONAL INFERENCE PROBLEMS

We showed above that, on low-dimensional examples, GATSBI does not perform as well as the best SBI methods. However, we expected that it would excel on problems on which GANs excel: high-dimensional and structured parameter spaces, in which suitably designed neural networks can be used as powerful classifiers. We apply GATSBI to two problems with ∼100-dimensional parameter spaces, one of which also has an implicit prior.

**Shallow Water Model**   The 1D shallow water equations describe the propagation of an initial disturbance across the surface of a shallow basin with an irregular depth profile and bottom friction (Figure 3A). They are relevant to oceanographers studying the dynamics on the continental sea shelf (Backhaus, 1983; Holton, 1973). We discretised the partial differential equations on a 100-point spatial grid, obtaining a simulator parameterised by the depth profile of the basin, $\theta \in \mathbb{R}^{100}$ (substantially higher than most previous SBI-applications, which generally are $\approx 10$ dim). The resulting amplitude of the waves evolving over a total of 100 time steps constitutes the 10k-dimensional raw observation from which we aim to infer the depth profile. The observation is taken into the Fourier domain, where both the real and imaginary parts receive additive noise and are concatenated, entering the inference pipeline as an array $x \in \mathbb{R}^{20k}$. Our task was to estimate $p(\theta|x)$, i.e., to infer the basin depth profile from a noisy Fourier transform of the surfaces waves. We trained GATSBI with 100k depth profiles sampled from the prior and the resulting surface wave simulations. For comparison, we trained NPE and NLE on the same data, using the same embedding net as the GATSBI discriminator to reduce the high dimensionality of the observations $x \in \mathbb{R}^{20k}$ to $\mathbb{R}^{100}$.

Samples from the GATSBI posterior captured the structure of the underlying ground-truth depth profile (Figure 3B, left).  In comparison, only NPE, but not NLE, captured the true shape of the depth profile (Figure 3C, D, left). In addition, the inferred posterior means for GATSBI and NPE, but not NLE, were correlated with the ground-truth parameters ($0.88 \pm 0.10$ for GATSBI and $0.91 \pm 0.09$ for NPE, mean $\pm$ standard deviation across 1000 different observations). Furthermore, while GATSBI posterior predictive samples did not follow the ground-truth observations as closely as NPE's, they captured the

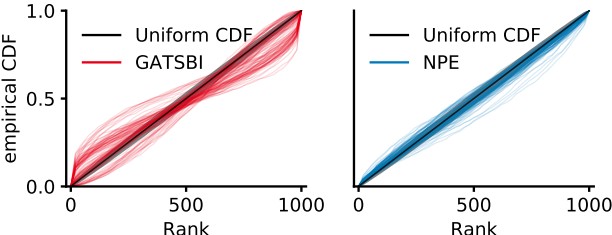

Figure 4: SBC results on shallow water model. Empirical cdf of SBC ranks, one line per posterior dimension (red, GATSBI; blue, NPE). In gray, region showing 99% of deviation expected under the ideal uniform cdf (black).

structure and were not as phase-shifted as NLE's (Figure 3 B-D). This is expected since wave-propagation speed depends strongly on depth profile, and thus incorrect inference of the depth parameters leads to temporal shifts between observed and inferred waves. Finally, simulation-based calibration (SBC) (Talts et al., 2020) shows that both NPE and GATSBI provide well-calibrated posteriors on this challenging, 100 dimensional inference problem (clearly, neither is perfectly calibrated, Figure 4). This realistic simulator example illustrates that GATSBI can learn posteriors with a dimensionality far from feasible for most SBI algorithms. It performs similar to NPE, previously reported to be a powerful approach on high-dimensional SBI problems (Lueckmann et al., 2021).

**Noisy Camera Model**   Finally, we demonstrate a further advantage of GATSBI over many SBI algorithms: GATSBI (like LFVI) can be applied in cases in which the prior distribution is only specified through an implicit model. We illustrate this by considering inference over images from a noisy and blurred camera model. We interpret the model as a black-box simulator. Thus, the clean images $\theta$ are samples from an implicit prior over images $\pi(\theta)$, and the simulated sensor activations are observations $x$. We can then recast the task of denoising the images as an inference problem (we note that our goal is *not* to advance the state-of-the-art for super-resolution algorithms, for which multiple powerful approaches are available, e.g. Kim et al., 2016; Zhang et al., 2018).

Since images are typically high-dimensional objects, learning a posterior over images is already a difficult problem for many SBI algorithms – either due to the implicit prior or the curse of dimensionality making MCMC sampling impracticable.  However, the implicit prior poses no challenge for GATSBI since it is trained only with samples.

We chose the EMNIST dataset (Cohen et al., 2017) with  800k 28×28-dimensional images as the implicit prior. This results in an inference problem in a 784-dimensional parameter space—much higher dimensionality than in typical SBI applications. The GATSBI generator was trained to produce clean images, given only the blurred image as input, while the discriminator was trained with clean images from the prior and the generator, and the corresponding blurred images. GATSBI indeed recovered crisp sources from blurred observations: samples from the GATSBI posterior given blurred observations accurately matched the underlying clean ground-truth images (see Figure 5). In contrast,

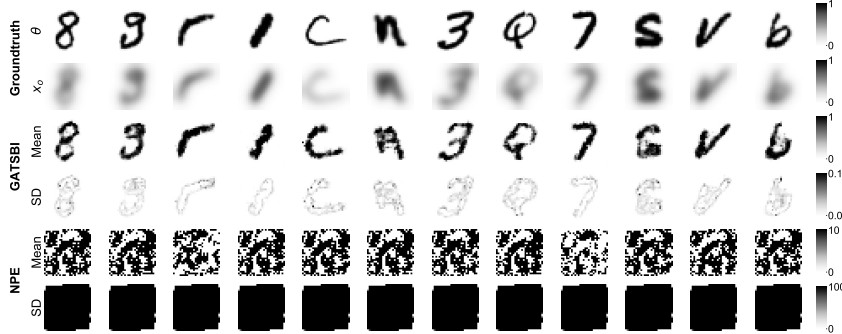

Figure 5: **Camera model inference.** Top: ground-truth parameter samples from the implicit prior and corresponding blurred camera model observations. Bottom: mean and standard deviation (SD) of inferred GATSBI and NPE posterior samples for matching observation from top.

NPE did not produce coherent posterior samples, and took substantially longer to train than GATSBI (NLE and NRE are not applicable due to the implicit prior).

## 4 DISCUSSION

We proposed a new method for simulation-based inference in stochastic models with intractable likelihoods. We used conditional GANs to fit a neural network functioning as an implicit density, allowing efficient, scalable sampling. We clarified the relationship between GANs and SBI, and showed how our method is related to adversarial VAEs that learn implicit posteriors over latent variables. We showed how GATSBI can be extended for sequential estimation, fine-tuning inference to a particular observation at the expense of amortisation. In contrast to conventional density-estimation based SBI-methods, GATSBI targets the *reverse* $D_{\mathrm{KL}}$—by thus extending both amortised and sequential SBI, we made it possible to explore the advantages of the forward and reverse $D_{\mathrm{KL}}$.

We found GATSBI to work adequately on two low-dimensional benchmark problems, though it did not outperform neural network-based methods. However, its real potential comes from the ability of GANs to learn generative models in high-dimensional problems. We demonstrated this by using GATSBI to infer a well-calibrated posterior over a 100-dimensional parameter space for the shallow water model, and a 784-dimensional parameter space for the camera model. This is far beyond what is typically attempted with SBI: indeed we found that GATSBI outperformed NPE on the camera model, and performed similarly to NPE and substantially outperformed NLE on the shallow water model (both SBI approaches were previously shown to be more powerful than others in high dimensions; Lueckmann et al., 2021). Moreover, we showed that GATSBI can learn posteriors in a model in which the prior is only defined implicitly (i.e., through a database of samples, not a parametric model).

Despite the advantages that GATSBI confers over other SBI algorithms in terms of flexibility and scalability, it comes with computational and algorithmic costs. GANs are notoriously difficult to train—they are highly sensitive to hyperparameters (as demonstrated in the two-moons example) and problems such as mode collapse are common. Moreover, training time is significantly higher than in other SBI algorithms. In addition, since GATSBI encodes an implicit posterior, i.e., a density from which we can sample but cannot evaluate, it yields posterior samples but not a parametric model. Thus, GATSBI is unlikely to be the method of choice for low-dimensional problems. Similarly, sequential GATSBI requires extensive hyperparameter tuning in order to produce improvements over amortised GATSBI, with the additional cost of training a classifier to approximate the correction factor. Hence, sequential GATSBI might be useful in applications where the computational cost of training is offset by having an expensive simulator. We note that GATSBI might benefit from recent improvements on GAN training speed and sample efficiency (Sauer et al., 2021).

Overall, we expect that GATSBI will be particularly impactful on problems in which the parameter space is high-dimensional and has GAN-friendly structure, e.g. images. Spatially structured models and data abound in the sciences, from climate modelling to ecology and economics. High-dimensional parameter regimes are common, but inaccessible for current SBI algorithms. Building on the strengths of GANs, we expect that GATSBI will help close this gap and open up new application-domains for SBI, and new directions for building SBI methods employing adversarial training.

## ACKNOWLEDGMENTS

We thank Kai Logemann for providing code for the shallow water model. We thank Michael Deistler, Marcel Nonnenmacher and Giacomo Bassetto for in-depth discussions; Auguste Schulz, Richard Gao, Artur Speiser and Janne Lappalainen for feedback on the manuscript and the reviewers at ICML 2021, NeurIPS 2021 and ICLR 2022 for insightful feedback. This work was funded by the German Research Foundation (DFG; Germany's Excellence Strategy MLCoE – EXC number 2064/1 PN 390727645; SPP 2041; SFB 1089 'Synaptic Microcircuits'), the German Federal Ministry of Education and Research (BMBF; project ADIMEM, FKZ 01IS18052 A-D; Tübingen AI Center, FKZ: 01IS18039A) and the Helmholtz AI initiative.

## ETHICS STATEMENT

Since this is a purely exploratory and theoretical work based on simulated data, we do not foresee major ethical concerns. However, we also note that highly realistic GANs have a potential of being exploited for applications with negative societal impacts e.g. deep fakes (Tolosana et al., 2020).

## REPRODUCIBILITY STATEMENT

To facilitate reproducibility, we provide detailed information about simulated data, model set up, training details and experimental results in the Appendix. Code implementing the method and experiments described in the manuscript is available at `https://github.com/mackelab/gatsbi`.

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

# APPENDIX

## A PROOFS

### A.1 PROOF OF CONVERGENCE FOR GATSBI

**Proposition 1.** *Given a fixed generator $f_\phi$, the discriminator $D_{\psi^*}(\theta, x)$ maximising equation 2 satisfies*

$$D_{\psi^*}(\theta, x) = \frac{p(\theta|x)}{p(\theta|x) + q_\phi(\theta|x)},$$

*and the corresponding loss function for the generator parameters is the Jensen-Shannon divergence (JSD) between the true and approximate posterior:*

$$L_{\psi^*}(\phi) = 2 \operatorname{JSD}(p(\theta|x) \,||\, q_\phi(\theta|x)) - \log 4$$

*Proof.* We start with equation 2. The proof proceeds as for Proposition 1 and 2 in Goodfellow et al. (2014). For convenience, we elide the arguments of the various quantities, so that $q_\phi$ denotes $q_\phi(\theta|x)$, $p$ denotes $p(\theta|x)$ and $D_\psi$ denotes $D_\psi(\theta, x)$.

$$
\begin{aligned}
L(\phi, \psi) &= \mathbb{E}_{p(x)}\Big[\mathbb{E}_p \log D_\psi + \mathbb{E}_{q_\phi} \log\big(1 - D_\psi\big)\Big] \\
&= \mathbb{E}_{p(x)}\Big[\int p \log D_\psi \, d\theta + \int q_\phi \log(1 - D_\psi) \, d\theta\Big] \\
&= \mathbb{E}_{p(x)}\Big[\int \big(p \log D_\psi + q_\phi \log(1 - D_\psi)\big) \, d\theta\Big].
\end{aligned}
$$

For any function $g(x) = a \log x + b \log(1-x)$, where $a, b \in \mathbb{R}^2 \setminus \{0, 0\}$ and $x \in (0, 1)$, the maximum of the function is at $x = a/a + b$. Hence, $L(\phi, \psi)$, for a fixed $\phi$, achieves it's maximum at:

$$D_{\psi^*} = \frac{p}{p + q_\phi}$$

Note that $p(x)$ drops out of the expression for $D_{\psi^*}$ since it is common to both terms in $L(\phi, \psi)$. Plugging this into equation 2 and dropping the expectation over $x$ without loss of generality:

$$
\begin{aligned}
L_{\psi^*}(\phi) &= \mathbb{E}_p \, \log \frac{p}{p + q_\phi} + \mathbb{E}_{q_\phi} \log \frac{q_\phi}{p + q_\phi} \\
&= \mathbb{E}_p \, \log \frac{2\,p}{p + q_\phi} + \mathbb{E}_{q_\phi} \log \frac{2\,q_\phi}{p + q_\phi} - \log 4 \\
&= 2 \operatorname{JSD}(p \,||\, q_\phi) - \log 4.
\end{aligned}
$$

The JSD is always non-negative, and zero only when $p(\theta|x) = q_\phi(\theta|x)$. Thus, the global minimum $L_{\psi^*}(\phi^*) = -\log 4$ is achieved only when the GAN generator converges to the ground-truth posterior $p(\theta|x)$. □

### A.2 PROOF FOR GATSBI MAXIMIZING ELBO LOSS

**Proposition 2.** *If $\phi^*$ and $\psi^*$ denote the Nash equilibrium of a min-max game defined by equation 2 then $\phi^*$ also maximises the evidence lower bound of a VAE with a fixed decoder i.e.,*

$$\phi^* = \arg\max_\phi L_E(\phi) \tag{8}$$

$$= \arg\max_\phi \mathbb{E}_{p(x)}\mathbb{E}_{q_\phi(\theta|x)} \left(\log \frac{\pi(\theta)}{q_\phi(\theta|x)} + \log p(x|\theta)\right) \tag{9}$$

*Proof.* The proposition is trivially true if $q_{\phi^*}(\theta|x) = p(\theta|x)$, i.e., $p(\theta|x)$ is the true minimum of both the JSD and $D_{KL}$. However, we here show that the equivalence holds even when $q_{\phi^*}(\theta|x)$ is *not*

the true posterior, e.g. if $q_\phi(\theta|x)$ belongs to a family of distributions that does not include the true posterior.

Given that $\phi^*$ and $\psi^*$ denote the Nash equilibrium in equation 2, we know from Proposition 1 that the optimal discriminator is given by

$$D_{\psi^*}(\theta, x) = \frac{p(\theta|x)}{p(\theta|x) + q_{\phi^*}(\theta|x)}.$$

To lighten notation, we elide the parameters of the networks and their arguments, denoting $D_{\psi^*}(\theta|x)$ as $D_{\psi^*}$, $q_{\phi^*}(\theta|x)$ as $q^*$, and so on. If we plug $D_{\psi^*}$ into Eq. 2, we have

$$\phi^* = \arg\min_\phi \; \mathbb{E}_{p(\theta|x)} \; \log \; D_{\psi^*} + \mathbb{E}_{q_\phi}\Big(\log\big(1 - D_{\psi^*}\big)\Big)$$
$$= \arg\min_\phi \mathbb{E}_{q_\phi} \; \log \big(1 - D_{\psi^*}\big) \tag{10}$$

Note that this is true *only* at the Nash equilibrium, where $D_{\psi^*}$ is a function of $q_{\phi^*}$ and *not* $q_\phi$. This allows us to drop the first term from the equation. In other words, if we switch out $q_{\phi^*}$ with any other $q_\phi$ in the expectation, equation 10 is no longer minimum w.r.t $\phi$, even though $D_{\psi^*}$ is optimal.

Let us define $D_\psi := \sigma(R_\psi)$, where $\sigma(\cdot) := 1/1 + \exp(-\cdot)$. Then from equation 3,

$$\sigma(R_{\psi^*}) = \frac{p(\theta|x)}{p(\theta|x) + q_{\phi^*}} \implies \frac{1}{1 + e^{-R_{\psi^*}}} = \frac{1}{1 + q_{\phi^*}/p(\theta|x)}$$

Comparing the l.h.s. and r.h.s. of the equation above, we get

$$R_{\psi^*} = \log \frac{p(\theta|x)}{q_{\phi^*}}. \tag{11}$$

Since both $\log$ and $\sigma$ are monotonically increasing functions, we also have from equation 10:

$$\phi^* = \arg\min_\phi \mathbb{E}_{q_\phi} \; \log \big(1 - \sigma(R^*)\big)$$
$$= \arg\min_\phi \mathbb{E}_{q_\phi} \big(1 - \sigma(R_{\psi^*})\big)$$
$$= \arg\max_\phi \mathbb{E}_{q_\phi} \; \sigma(R_{\psi^*})$$
$$= \arg\max_\phi \mathbb{E}_{q_\phi}(R_{\psi^*}) \tag{12}$$

In other words, $q_{\phi^*}$ maximises the function $\mathbb{E}_{q_\phi}(R_{\psi^*})$. Now, to prove equation 9, we need to show that $L_E(\phi) < L_E(\phi^*) \; \forall \; \phi \neq \phi^*$.

$$L_E(\phi) = \mathbb{E}_{p(x)}\mathbb{E}_{q_\phi} \left(\log \pi(\theta) - \log q_\phi + \log p(x|\theta)\right)$$
$$= \mathbb{E}_{p(x)}\mathbb{E}_{q_\phi}\left(\log \frac{p(\theta|x)}{q_\phi} + \log \frac{p(x|\theta)\pi(\theta)}{p(\theta|x)}\right)$$
$$= \mathbb{E}_{p(x)}\mathbb{E}_{q_\phi}\left(\log \frac{p(\theta|x)}{q_\phi} + \log p(x)\right)$$
$$= \mathbb{E}_{p(x)}\mathbb{E}_{q_\phi}\left(\log \frac{p(\theta|x)}{q_{\phi^*}} + \log p(x)\right) - \mathbb{E}_{p(x)}(D_{KL}(q_\phi||q_{\phi^*}))$$
$$< \; \mathbb{E}_{p(x)}\mathbb{E}_{q_\phi}\left(\log \frac{p(\theta|x)}{q_{\phi^*}} + \log p(x)\right)$$
$$= \mathbb{E}_{p(x)}\mathbb{E}_{q_\phi}\left(R_{\psi^*} + \log p(x)\right) \qquad \text{from equation 11}$$
$$< \; \mathbb{E}_{p(x)}\mathbb{E}_{q_{\phi^*}}\left(R_{\psi^*}\right) + \mathbb{E}_{p(x)}\mathbb{E}_{q_\phi} \log p(x) \qquad \text{from equation 12}$$
$$= \mathbb{E}_{p(x)}\mathbb{E}_{q_{\phi^*}}\left(R_{\psi^*} + \log p(x)\right) + \mathbb{E}_{p(x)}\mathbb{E}_{q_\phi} \log p(x) - \mathbb{E}_{p(x)}\mathbb{E}_{q_{\phi^*}} \log p(x)$$
$$= \mathbb{E}_{p(x)}\mathbb{E}_{q_{\phi^*}}\left(\log \frac{p(\theta|x)}{q_{\phi^*}} + \log p(x)\right)$$
$$= L_E(\phi^*)$$
$$\implies L_E(\phi) < L_E(\phi^*)$$

Hence, the approximate posterior obtained by optimising the GAN objective also maximises the evidence lower bound of the corresponding VAE. $\qquad \square$

### A.3 CONNECTION BETWEEN LFVI, DEEP POSTERIOR SAMPLING AND GATSBI

Adversarial inference approaches maximise the Evidence Lower Bound (ELBO) equation 6 to train a VAE, and use a discriminator to approximate intractable ratios of densities in the loss (see Table 1). Likelihood-free variational inference (Tran et al., 2017, LFVI) is one such method.

Table 1: Comparison of adversarial inference algorithms: BiGAN (Donahue et al., 2019), ALI (Dumoulin et al., 2016), AAE (Makhzani et al., 2015), AVB (Mescheder et al., 2017), and LFVI (Tran et al., 2017).

| ALGORITHM | DISCRIMINATOR RATIO | GENERATOR LOSS FUNCTION |
|---|---|---|
| BIGAN, ALI | $p_\alpha(x\|u)p(u)/q_\phi(u\|x)p(x)$ | $\mathrm{JSD}(p_\alpha(u,x)\|\|q_\phi(u,x))$ |
| AAE | $p(u)/q_\phi(u)$ | $\mathrm{JSD}(p(u)\|\|q_\phi(u))$ |
| AVB | $p(u)p(x)/q_\phi(u\|x)p(x)$ | $D_{\mathrm{KL}}(q_\phi(u\|x)\|\|p(u))$ |
| LFVI | $p(u\|x)p(x))/q_\phi(u\|x)p(x)$ | $D_{\mathrm{KL}}(q_\phi(u\|x)\|\|p(u\|x))$ |
| GATSBI | $p(u\|x)p(x)/q_\phi(u\|x)p(x)$ | $\mathrm{JSD}(p(u\|x)\|\|q_\phi(u\|x))$ |

In the most general formulation, LFVI learns a posterior over both latents $u$ and global parameters $\beta$ which are latents shared across multiple observations i.e., $q_\phi(z, \beta|x)$ and maximises the ELBO given by

$$L_{\mathrm{LF}}(\phi) = \mathbb{E}_{q_\phi(\beta)} \log \frac{p(\beta)}{q_\phi(\beta)} + \mathbb{E}_{q_\phi(u|x)p'(x)} \log \frac{p(x|u)p(u)}{q_\phi(u|x)p'(x)} + \mathrm{const.} \qquad (13)$$

where $p'(x)$,[3] an *empirical distribution* over observations, and *not* necessarily $p(x)$, the marginal likelihood of the simulator. A discriminator, $D_\psi(x, u)$,[4] is trained with the cross-entropy loss to approximate the intractable ratio $\frac{p(x|u)p(u)}{q_\phi(u|x)p'(x)}$ in the second term. Using the nomenclature from Huszár (2017), we note that $D_\psi$ is *joint-contrastive*: it simultaneously discriminates between tuples $(x, u) \sim p(x|u)p(u)$ and $(\hat{x}, \hat{u}) \sim q_\phi(\hat{u}|\hat{x})p'(\hat{x})$. GATSBI, by contrast, is *prior-contrastive*: its discriminator only discriminates between parameters $\theta$, given a fixed $x$.

However, when $p'(x) = p(x)$ and $\beta$ is constant, the LFVI discriminator becomes prior-contrastive, equation 13 is a function of both the discriminator parameters $\psi$ and generator parameters $\phi$, and it differs from GATSBI only by a single term i.e., from equation 2 and equation 13 and ignoring the constant:

$$L_{\mathrm{GATSBI}}(\phi, \psi) = -L_{\mathrm{LF}}(\phi, \psi) + \mathbb{E}_{q_\phi(u|x)p(x)} \log D_\psi(x, u) \qquad (14)$$

The second term corresponds to the non-saturating GAN loss (Goodfellow et al., 2014). In this

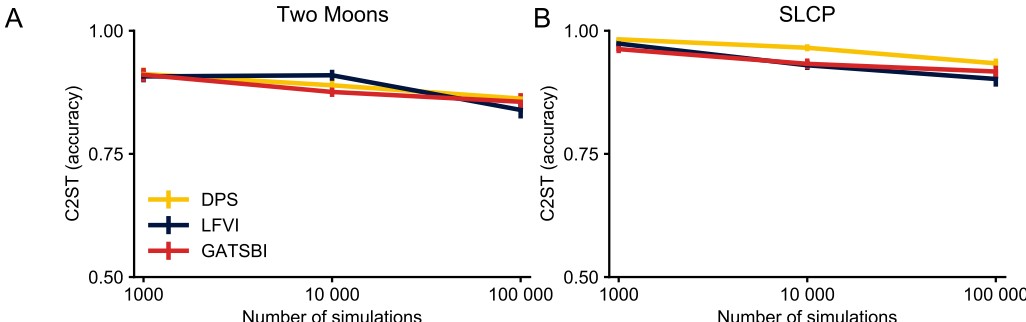

Figure 6: **GATSBI, LFVI and Deep Poseterior Sampling (DPS) on benchmark problems.** Mean C2ST score ($\pm$ standard error of the mean) for 10 observations each.  A. On Two Moons, the C2ST scores for GATSBI (red), LFVI (navy) and Deep Posterior Sampling (DPS, yellow) are qualitatively similar across all simulation budgets  B. On SLCP, DPS is slightly worse than LFVI and GATSBI.

---

[3] In Tran et al. (2017), $p'(x)$ is denoted $q(x)$

[4] denoted $r_\psi(x, u)$ in Tran et al. (2017)

setting, with an optimal discriminator, GATSBI minimises a JSD, whereas LFVI minimises the reverse $D_{\mathrm{KL}}$.

Adler and Öktem (2018) introduce Deep Posterior Sampling which also implements an adversarial algorithm for posterior estimation. The set-up is similar to GATSBI, but GANs are trained using a Wasserstein loss as in Arjovsky et al. (2017). The Wasserstein loss imposes stronger conditions on the GAN networks in order for the generator to recover the target distribution, i.e., the discriminator has to be 1-Lipschitz and the generator K-Lipschitz (Arjovsky et al., 2017; Qi, 2018). However, the JSD loss function allowed us to outline GATSBI's connection to adversarial VAEs and subsequently its advantages for SBI (see Sec. 2.4, Suppl. Sec. A.2 and the discussion above). Whether this would also be possible with the Wasserstein loss remains a subject for future work. Nevertheless, the sequential extension of GATSBI using the energy-based correction (see Sec. 2.5 and Supp. Sec. A.4) could in principle also be used with the Wasserstein metric.

Mescheder et al. (2018) state that the WGAN converges only when the discriminator minimizes the Wasserstein metric at every step, which does not happen in practice. Fedus et al. (2017) argue that a GAN generator does not necessarily minimise a JSD at *every* update step since the discriminator is optimal only in the limit of infinite data. Hence, neither asymptotic property can be used to reason about GAN behaviour in practice. As a consequence, it is difficult to predict the conditions under which LFVI, or Deep Posterior Sampling would outperform GATSBI, or vice-versa. Nevertheless, given the same networks and hyperperparameters (with slight modifications to the discriminator for Deep Posterior Sampling, see App. D.1 for details), we found empirically that LFVI, Deep Posterior Sampling and GATSBI are qualitatively similar on Two Moons, and that Deep Posterior Sampling is slightly worse that the other two algorithms on SLCP: i.e., there is no advantage to using one algorithm over the other on the problems investigated (see Fig. 6).

### A.4 SEQUENTIAL GATSBI

For many SBI applications, the density estimator is only required to generate good posterior samples for a particular experimentally observed data $x_{\mathrm{o}}$. This can be achieved by training the density estimator using samples $\theta$ from a proposal prior $\tilde{\pi}(\theta)$ instead of the prior $\pi(\theta)$. The proposal prior ideally produces parameters $\theta$ that are localised around the modes of $p(\theta|x_{\mathrm{o}})$ and can guide the density estimator towards inferring a posterior that is accurate for $x \approx x_{\mathrm{o}}$. If we replace the true prior $\pi(\theta)$ with a proposal prior $\tilde{\pi}(\theta) = q_\phi(\theta|x_{\mathrm{o}})$, i.e., the posterior estimated by GATSBI, and sample from the respective distributions

$$\theta, x \sim \tilde{\pi}(\theta)p(x|\theta),$$

the corresponding GAN loss (from equation equation 2) is:

$$\tilde{L}(\phi, \psi) = \mathbb{E}_{\tilde{\pi}(\theta)p(x|\theta)p(z)} \left[ \log \mathrm{D}_\psi(\theta, x) + \log(1 - \mathrm{D}_\psi(f_\phi(z, x), x)) \right] \tag{15}$$

$$= \mathbb{E}_{\tilde{p}(\theta|x)\tilde{p}(x)} \log \mathrm{D}_\psi(\theta, x) + \mathbb{E}_{q_\phi(\theta|x)\tilde{p}(x)} \log(1 - \mathrm{D}_\psi(\theta, x)). \tag{16}$$

This loss would allow us to obtain a generator that produces samples $\theta$ that are likely to generate outputs $x$ close to $x_o$, when plugged into the simulator. However, Proposition 1 in Appendix A.1 shows that this loss would result in $q_\phi(\theta|x)$ converging to the proposal posterior $\tilde{p}(\theta|x)$ rather than the ground-truth posterior $p(\theta|x)$. In order to learn a conditional density that is accurate for $x \approx x_{\mathrm{o}}$ but nevertheless converges to the correct posterior, we need to correct the approximate posterior for the bias due to the proposal prior. We outline three different approaches to this correction step:

**Using energy-based GANs** Although it is possible to use correction factors directly in the GATSBI loss function, as we outline in the next section, these corrections can lead to unstable training (Papamakarios et al., 2019). Here, we outline an approach in which we train on samples from the proposal prior without explicitly introducing correction factors into the loss function. Instead, we change the setup of the generator to produce 'corrected' samples, which are then used to compute the usual cross-entropy loss, and finally we train the discriminator and generator.

Let us start by introducing the correction factor $\omega(\theta, x) = \frac{\pi(\theta)}{\tilde{\pi}(\theta)} \frac{\tilde{p}(x)}{p(x)}$, such that

$$p(\theta|x) = \tilde{p}(\theta|x) \, \omega(\theta, x). \tag{17}$$

In the original formulation of GATSBI, sampling from $q_\phi(\theta|x)$ entails sampling latents $z \sim p(z)$, and transforming them by a deterministic function $f_\phi(x, z)$ to get parameters $\theta$ (see equation equation 1).

Following the energy-based GAN formulation (Azadi et al., 2019; Che et al., 2020), we define an *intermediate* latent distribution $p_t(z)$:

$$p_t(z) = p(z)(\omega(f_\phi(x, z), x))^{-1}. \tag{18}$$

$p_t(z)$ is the distribution of latent variables that, when passed through the function $f_\phi(x, z)$, are most likely to produce samples from the *approximate proposal posterior* $\tilde{q}_\phi(\theta|x) = q_\phi(\theta|x)(\omega(\theta, x))^{-1}$. For GANs, $p(z)$ is typically a tractable distribution whose likelihood can be computed, and from which one can sample, and thus, we can use MCMC or rejection sampling to also sample from $p_t(z)$ (see Appendix D.1 for details). The resulting loss function is:

$$\tilde{L}(\phi, \psi) = \mathbb{E}_{\tilde{p}(\theta|x)\tilde{p}(x)} \log D_\psi(\theta, x) + \mathbb{E}_{p_t(z)\tilde{p}(x)} \log(1 - D_\psi(f_\phi(x, z), x)) = \mathbb{E}_{\tilde{p}(x)}[L_1 + L_2]. \tag{19}$$

Note that there are no explicit correction factors in the loss.

We now show that optimising the loss function equation 19 leads to the generator converging to the correct posterior distribution. Let us first focus on the second term $L_2$:

$$
\begin{aligned}
L_2 &= \mathbb{E}_{p_t(z)} \log(1 - D_\psi(f_\phi(x, z), x)) \\
&= \int p_t(z) \log(1 - D_\psi(f_\phi(x, z), x)) \\
&= \int p(z) \, (\omega(f_\phi(x, z), x))^{-1} \, \log(1 - D_\psi(f_\phi(x, z), x)) \qquad \text{from equation 18} \\
&= \int q_\phi(\theta|x) \, (\omega(\theta, x))^{-1} \, \log(1 - D_\psi(\theta, x)) \qquad\qquad \text{reparam. trick} \\
&= \int \tilde{q}_\phi(\theta|x) \, \log(1 - D_\psi(\theta, x)) \\
&= \mathbb{E}_{\tilde{q}_\phi(\theta|x)} \log(1 - D_\psi(\theta, x)).
\end{aligned}
$$

Thus, from Proposition 1, we can conclude that by optimising the loss function equation 19, $\tilde{q}_\phi(\theta|x) \to \tilde{p}(\theta|x)$. This implies that the generator network, which represents $q_\phi(\theta|x)$, converges to $p(\theta|x)$, since both the approximate and target proposal posteriors are related respectively to the approximate and true posteriors by the same multiplicative factor $\omega(\theta, x)$. Note that the generator is more accurate in estimating the posterior given $x_o$ (or $x \approx x_o$), i.e., $p(\theta|x_o)$ than given $x$ far from $x_o$, since it is trained on samples from the proposal prior.

In practice, this scheme does produce improvements in the learned posterior. However, it is computationally expensive, because *every update* to the generator and discriminator requires a round of MCMC or rejection sampling to obtain the 'corrected' samples. Moreover, if we use the generator from the previous round as the proposal prior in the next round, we need to train a classifier to approximate $\omega(\theta, x)$ at every round.[5] Finally, this approach has additional hyperparameters that need to be tuned during GAN training, which could make it prohibitively difficult to use for most applications.

Below, we outline theoretical arguments for two additional approaches, although we only provide empirical results for the second approach.

**Using importance weights** Lueckmann et al. (2017) solve the problem of bias from using a proposal prior by introducing importance weights in their loss function. One can use the same trick for GATSBI, by introducing the importance weights $\omega(\theta, x) = \frac{\pi(\theta)}{\tilde{\pi}(\theta)} \frac{\tilde{p}(x)}{p(x)}$ into the loss defined in equation 16:

$$\tilde{L}(\phi, \psi) = \mathbb{E}_{\tilde{\pi}(\theta)p(x|\theta)p(z)} \left[ \omega(\theta, x) \log D_\psi(\theta, x) + \log(1 - D_\psi(f_\phi(z, x), x)) \right] = L_1 + L_2. \tag{20}$$

---

[5]Note that the correction factor could be computed in closed form if we had a generator with an evaluable density: we would not have to train a classifier to approximate it.

Optimising this loss allows $q_\phi(\theta|x)$ to converge to the true posterior $p(\theta|x)$. Let us focus on the first term $L_1$:

$$
\begin{aligned}
L_1 &= \mathbb{E}_{\tilde{p}(\theta|x)\tilde{p}(x)}\omega(\theta,x)\log \mathrm{D}_\psi(\theta,x) \\
&= \int_x \int_\theta \tilde{p}(x)\tilde{p}(\theta|x)\frac{\pi(\theta)}{\tilde{\pi}(\theta)}\frac{\tilde{p}(x)}{p(x)}\log \mathrm{D}_\psi(\theta,x) \\
&= \int_x \int_\theta p(x|\theta)\tilde{\pi}(\theta)\frac{\pi(\theta)}{\tilde{\pi}(\theta)}\frac{\tilde{p}(x)}{p(x)}\log \mathrm{D}_\psi(\theta,x) \\
&= \int_x \int_\theta p(x|\theta)\pi(\theta)\frac{\tilde{p}(x)}{p(x)}\log \mathrm{D}_\psi(\theta,x) \\
&= \int_x \int_\theta p(x)p(\theta|x)\frac{\tilde{p}(x)}{p(x)}\log \mathrm{D}_\psi(\theta,x) \\
&= \int_x \int_\theta \tilde{p}(x)p(\theta|x)\log \mathrm{D}_\psi(\theta,x) \\
&= \mathbb{E}_{\tilde{p}(x)p(\theta|x)}\log \mathrm{D}_\psi(\theta,x).
\end{aligned}
$$

Thus, from Proposition 1, we can conclude that by optimising the loss function equation 20, the generator $q_\phi(\theta|x)$ converges to the true posterior. However, the importance-weight correction could lead to high-variance gradients (Papamakarios et al., 2019). This would be particularly problematic for GANs, where the loss landscape for each network is modified with updates to its adversary, and there is no well-defined optimum. High-variance gradients could cause training to take longer or even prevent it from converging altogether.

**Using inverse importance weights** Since using importance weights in the loss can lead to high-variance gradients, we could instead consider using the inverse of the importance weights $(\omega(\theta,x))^{-1} = \frac{\tilde{\pi}(\theta)}{\pi(\theta)}\frac{p(x)}{\tilde{p}(x)}$ in the second term in equation 16:

$$
\tilde{L}(\phi,\psi) = \mathbb{E}_{\tilde{p}(\theta|x)\tilde{p}(x)}\log \mathrm{D}_\psi(\theta,x) + \mathbb{E}_{q_\phi(\theta|x)\tilde{p}(x)}(\omega(\theta,x))^{-1}\log(1-\mathrm{D}_\psi(\theta,x)) = L_1 + L_2.
\tag{21}
$$

Optimising $\tilde{L}(\phi,\psi)$ from equation 21 will result in the generator approximating the true posterior at convergence. Focusing on the second term of the loss function $L_2$:

$$
\begin{aligned}
L_2 &= \mathbb{E}_{q_\phi(\theta|x)\tilde{p}(x)}(\omega(\theta,x))^{-1}\log(1-\mathrm{D}_\psi(\theta,x)) \\
&= \iint \tilde{p}(x)q_\phi(\theta|x)\frac{\tilde{\pi}(\theta)}{\pi(\theta)}\frac{p(x)}{\tilde{p}(x)}\log(1-\mathrm{D}_\psi(\theta,x)) \\
&= \iint \tilde{p}(x)\tilde{q}_\phi(\theta|x)\log(1-\mathrm{D}_\psi(\theta,x)) \\
&= \mathbb{E}_{\tilde{p}(x)\tilde{q}_\phi(\theta|x)}\log(1-\mathrm{D}_\psi(\theta,x)).
\end{aligned}
$$

Thus, from Proposition 1, we can conclude that by optimising the loss function equation 21, $\tilde{q}_\phi(\theta|x)$ converges to $\tilde{p}(\theta|x)$. Since $\tilde{q}_\phi(\theta|x)$ differs from $q_\phi(\theta|x)$ by the same factor as $\tilde{p}(\theta|x)$ from $p(\theta|x)$, i.e., $(\omega(\theta,x))^{-1}$ (see equation equation 17), this implies that $q_\phi(\theta|x) \to p(\theta|x)$.

# B  TRAINING ALGORITHMS

---

**Algorithm 1** GATSBI

---

Input : prior $\pi(\theta)$, simulator $p(x|\theta)$, generator $f_\phi$, discriminator $\mathrm{D}_\psi$, learning rate $\lambda$
Output: Trained GAN networks $f_{\phi^*}$ and $\mathrm{D}_{\psi^*}$
$\Theta = \{\theta_1, \theta_2, \ldots, \theta_n\} \overset{\text{i.i.d}}{\sim} \pi(\theta)$
$\mathbf{X} = \{x_1, x_2, \ldots, x_n\} \sim p(x_i|\theta_i)$
**while** not converged **do**
    **for** discriminator iterations **do**
        Sample mini-batch $\mathbf{X}_d, \Theta_d$ from $\mathbf{X}, \Theta$
        $\mathbf{Z} \sim p(z), \hat{\Theta}_d = f_\phi(\mathbf{Z}, \mathbf{X}_d)$
        $L = \sum_{\mathbf{X}_d} \left( \sum_{\Theta_d} \log \mathrm{D}_\psi(\Theta_d, \mathbf{X}_d) + \sum_{\hat{\Theta}_d} \log(1 - \mathrm{D}_\psi(\hat{\Theta}_d, \mathbf{X}_d)) \right)$
        $\psi \leftarrow \psi + \lambda \nabla_\psi L$
    **end for**
    **for** generator iterations **do**
        Sample mini-batch $\mathbf{X}_g, \Theta_g$ from $\mathbf{X}, \Theta$
        $\mathbf{Z} \sim p(z), \hat{\Theta}_g = f_\phi(\mathbf{Z}, \mathbf{X}_g)$
        $L = -\sum_{\mathbf{X}_g} \sum_{\hat{\Theta}_g} \log(1 - \mathrm{D}_\psi(\hat{\Theta}_g, \mathbf{X}_g))$
        $\phi \leftarrow \phi + \lambda \nabla_\phi L$
    **end for**
**end while**

---

**Algorithm 2** Sequential GATSBI with energy-based correction

---

Input: $\pi(\theta)$, simulator $p(x|\theta)$, classifier $\omega = 1$, observation $x_\mathrm{o}$, $f_\phi$, $\mathrm{D}_\psi$, learning rate $\lambda$
Output: Trained GAN networks $f_{\phi^*}$ and $\mathrm{D}_{\psi^*}$
**for** $i = 1 \cdots$ number of rounds **do**
    $\Theta_i = \{\theta_1, \theta_2, \ldots, \theta_n\} \overset{\text{i.i.d}}{\sim} \pi(\theta)$
    $\mathbf{X}_i = \{x_1, x_2, \ldots, x_n\} \sim p(x|\theta)$
    **if** $i > 1$ **then**:
        $\omega_\theta \leftarrow \max_{\omega_\theta} \log \sigma(\omega_\theta(\Theta_0)) + \log(1 - \sigma(\omega_\theta(\Theta_i)))$
        $\omega_x \leftarrow \max_{\omega_x} \log \sigma(\omega_x(\mathbf{X}_0)) + \log(1 - \sigma(\omega_x(\mathbf{X}_i)))$
        $\omega = \frac{\omega_\theta}{\omega_x}$
    **end if**
    **while** not converged **do**
        **for** discriminator iterations **do**
            Sample mini-batch $\mathbf{X}_d, \Theta_d$ from $\mathbf{X}_i, \Theta_i$
            $\mathbf{Z} \sim p_t(\mathbf{Z}) = p(\mathbf{Z})(\omega(f_\phi(\mathbf{Z}, \mathbf{X}_d), \mathbf{X}_d))^{-1}$
            $\hat{\Theta}_d = f_\phi(\mathbf{Z}, \mathbf{X}_d)$
            $L = \sum_{\mathbf{X}_d} (\sum_{\Theta_d} \log \mathrm{D}_\psi(\Theta_d, \mathbf{X}_d) + \sum_{\hat{\Theta}_d} \log(1 - \mathrm{D}_\psi(\hat{\Theta}_d, \mathbf{X}_d)))$
            $\psi \leftarrow \psi + \lambda \nabla_\psi L$
        **end for**
        **for** generator iterations **do**
            Sample mini-batch $\mathbf{X}_g, \Theta_g$ from $\mathbf{X}_i, \Theta_i$
            $\mathbf{Z} \sim p_t(\mathbf{Z}) = p(\mathbf{Z})(\omega(f_\phi(\mathbf{Z}, \mathbf{X}_g), \mathbf{X}_g))^{-1}$
            $\hat{\Theta}_g = f_\phi(\mathbf{Z}, \mathbf{X}_g)$
            $L = -\sum_{\mathbf{X}_g} \sum_{\hat{\Theta}_g} \log(1 - \mathrm{D}_\psi(\hat{\Theta}_g, \mathbf{X}_g))$
            $\phi \leftarrow \phi + \lambda \nabla_\phi L$
        **end for**
    **end while**
    $\pi(\theta) = f_\phi(\theta; x_\mathrm{o})$
**end for**

---

---

**Algorithm 3** Sequential GATSBI with inverse importance weights

---

Input: $\pi(\theta)$, simulator $p(x|\theta)$, classifier $\omega = 1$, observation $x_{\mathrm{o}}$, $f_\phi$, $\mathrm{D}_\psi$, learning rate $\lambda$
Output: Trained GAN networks $f_{\phi^*}$ and $\mathrm{D}_{\psi^*}$

**for** $i = 1 \cdots$ number of rounds **do** $\Theta_i = \{\theta_1, \theta_2, \ldots, \theta_n\} \overset{\text{i.i.d}}{\sim} \pi(\theta)$ $\mathbf{X}_i = \{x_1, x_2, \ldots, x_n\} \sim p(x|\theta)$
    **if** $i > 1$ **then**:
        $\omega_\theta \leftarrow \max\limits_{\omega_\theta} \log \sigma(\omega_\theta(\Theta_0)) + \log(1 - \sigma(\omega_\theta(\Theta_i)))$
        $\omega_x \leftarrow \max\limits_{\omega_x} \log \sigma(\omega_x(\mathbf{X}_0)) + \log(1 - \sigma(\omega_x(\mathbf{X}_i)))$
        $\omega = \frac{\omega_\theta}{\omega_x}$
    **end if**
    **while** not converged **do**
        **for** discriminator iterations **do**
            Sample mini-batch $\mathbf{X}_d$, $\Theta_d$ from $\mathbf{X}_i$, $\Theta_i$
            $\mathbf{Z} \sim p(\mathbf{Z})$
            $\hat{\Theta}_d = f_\phi(\mathbf{Z}, \mathbf{X}_d)$
            $L = \sum_{\mathbf{X}_d}(\sum_{\Theta_d} \log \mathrm{D}_\psi(\Theta_d, \mathbf{X}_d) + \sum_{\hat{\Theta}_d}(\omega(\hat{\Theta}_d, \mathbf{X}_d))^{-1} \log(1 - \mathrm{D}_\psi(\hat{\Theta}_d, \mathbf{X}_d)))$
            $\psi \leftarrow \psi + \lambda \nabla_\psi L$
        **end for**
        **for** generator iterations **do**
            Sample mini-batch $\mathbf{X}_g$, $\Theta_g$ from $\mathbf{X}_i$, $\Theta_i$
            $\mathbf{Z} \sim p(\mathbf{Z})$
            $\hat{\Theta}_g = f_\phi(\mathbf{Z}, \mathbf{X}_g)$
            $L = -\sum_{\mathbf{X}_g} \sum_{\hat{\Theta}_g}(\omega(\hat{\Theta}_g, \mathbf{X}_g))^{-1} \log(1 - \mathrm{D}_\psi(\hat{\Theta}_g, \mathbf{X}_g))$
            $\phi \leftarrow \phi + \lambda \nabla_\phi L$
        **end for**
    **end while**
    $\pi(\theta) = f_\phi(\theta; x_{\mathrm{o}})$
**end for**

---

# C   ADDITIONAL RESULTS

## C.1   POSTERIORS FOR BENCHMARK PROBLEMS

We show posterior plots for the benchmark problems: SLCP (Fig. 7) and Two Moons (Fig. 8). In both figures, panels on the diagonal display the histograms for each parameter, while the off-diagonal panels show pairwise posterior marginals, i.e., 2D histograms for pairs of parameters, marginalised over the remaining parameter dimensions.

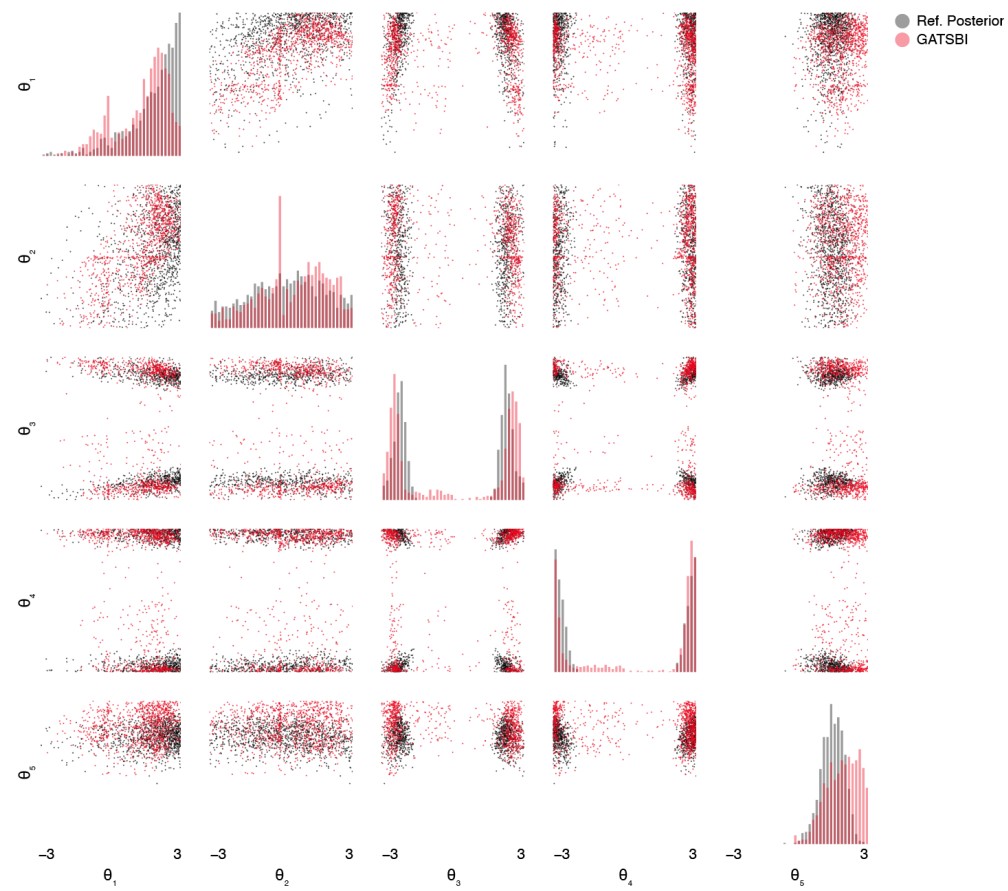

Figure 7: Inference for one test observation of the SLCP problem. Posterior samples for GATSBI trained on 100k simulations (red), and reference posterior samples (black). The GATSBI posterior samples cover well the disjoint modes of the posterior, although GATSBI sometimes produces samples in regions of low density in the reference posterior.

## C.2   SEQUENTIAL GATSBI

We found that sequential GATSBI with the energy-based correction produced a modest improvement over amortised GATSBI for the Two Moons model with 1k and 10k simulation budgets, and no improvement at all with 100k (see Fig. 9). The inverse importance weights correction did not produce an improvement for any simulation budget. Sequential GATSBI performance was also sensitive to hyperparameter settings and network initialisation. We hypothesise that further improvement is possible with better hyperparameter or network architecture tuning.

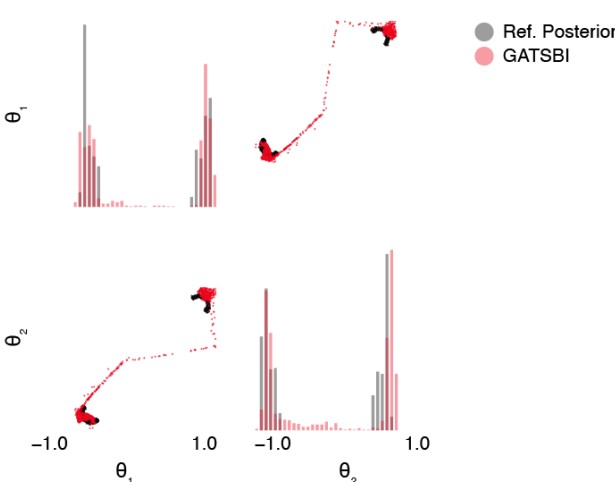

Figure 8: Inference for one test observation of the Two Moons problem. Posterior samples for GATSBI trained on 100k simulations (red), and reference posterior samples (black). GATSBI captures the global bi-modal structure in the reference posterior, but not the local crescent shape. It also generates some samples in regions of low density in the reference posterior.

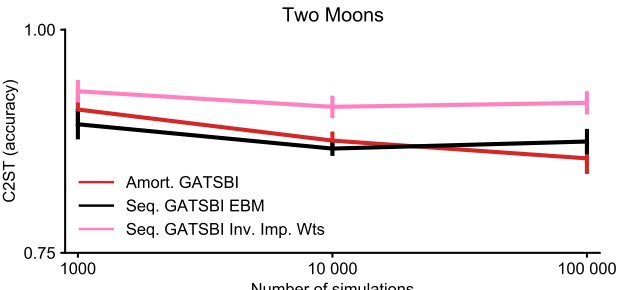

Figure 9: Sequential GATSBI performance for the Two Moons Model. The energy-based correction (EBM) results in a slight improvement over amortised GATSBI for 1k and 10k simulations, but the inverse importance weights correction does not.

# D IMPLEMENTATION DETAILS

All networks and training algorithms were implemented in PyTorch (Paszke et al., 2019). We used Weights and Biases (Biewald, 2020) to log experiments with different hyperparameter sets and applications. We ran the high-dimensional experiments (camera model and shallow water model) on Tesla-V100 GPUs: the shallow water model required training to be parallelised across 2 GPUs at a time, and took about 4 days to converge and about 1.5 days for the camera model on one Tesla V100. We used RTX-2080Tis for the benchmark problems: the amortised GATSBI runs lasted a maximum of 1.5 days for the 100k budget; the sequential GATSBI runs took longer with the maximum being 8 days for the energy-based correction with a budget of 100k. On similar resources, NPE took about 1 day to train on the shallow water model, and 3 weeks and 2 days to train on the camera model. NPE took about 10 min for 100k simulations on both benchmark problems. SMC-ABC and rejection-ABC both took about 6s on the benchmark problems with a budget of 100k.

## D.1 SIMPLE-LIKELIHOOD COMPLEX POSTERIOR (SLCP) AND TWO-MOONS

**Prior and simulator** For details of the prior and simulator, we refer to Lueckmann et al. (2021).

**GAN architecture** The generator was a 5-layer MLP with 128 hidden units in the first four layers. The final layer had output features equal to the parameter dimension of the problem. A leaky ReLU nonlinearity with slope 0.1 followed each layer. A noise vector sampled from a standard normal distribution (two-dimensional for Two Moons, and 5-dimensional for SLCP) was injected at the fourth layer of the generator. The generator received the observations as input to the first layer, used the first three layers to embed the observation, and multiplied the embedding with the injected noise at the fourth layer, which was then passed through the subsequent layers to produce an output of the same dimensions as the parameters. The discriminator was a 6-layer MLP with 2048 hidden units in the first five layers and the final layer returned a scalar output. A leaky ReLU nonlinearity with slope 0.1 followed each layer, except for the last, which was followed by a sigmoid nonlinearity. The discriminator received both the observations and the parameters sampled alternatively from the generator and the prior as input. Observations and parameters were concatenated and passed through the six layers of the network. For Deep Posterior Sampling, the discriminator did not have the final sigmoid layer.

**Training details** The generator and discriminator were trained in parallel for 1k, 10k and 100k simulations, with a batch size = $\min(10\%$ of the simulation budget, 1000). For each simulation budget, 100 samples were held out for validation. We used 10 discriminator updates for 1k and 10k simulation budgets, and 100 discriminator updates for the 100k simulation budget, per generator update. Note that the increase in discriminator updates for 100k simulations is intended to compensate for the reduced relative batch size i.e., 1000 batches = 0.01%. The networks were optimised with the cross-entropy loss. We used the Adam optimiser (Kingma and Ba, 2015) with learning rate=0.0001, $\beta_1$=0.9 and $\beta_2$=0.99 for both networks. We trained the networks for 10k, 20k and 20k epochs for the three simulation budgets respectively. To ensure stable training, we used spectral normalisation (Miyato et al., 2018) for the discriminator network weights. For the comparison with LFVI, we kept the architecture and hyperparameters the same as for GATSBI, but trained the generator to minimise $L(\phi) = \mathbb{E}_{q_\phi(\theta|x)}[\log \frac{1-D_\psi(\theta,x)}{D_\psi(\theta,x)}]$. Similarly, for Deep Posterior Sampling, we kept the same architecture (minus the final sigmoid layer for the discriminator) and hyperparameters, but trained the generator and discriminator on the Wasserstein loss: $L(\phi,\psi) = \mathbb{E}_{p(\theta|x)} D_\psi(\theta) - \mathbb{E}_{q_\phi(\theta|x)} D_\psi(\theta)$.

**Optimised hyperparameters for Two Moons model** The generator was a 2-layer MLP with 128 hidden units in the first layer and 2 output features in the second layer (same as the parameter dimension). Each layer was followed by a leaky ReLU nonlinearity with slope 0.1. Two-dimensional white noise was injected into the second layer, after it was multiplied with the output of the first layer. The discriminator was a 4-layer MLP with 2048 hidden units in the first 3 layers each followed by a leaky ReLU nonlinearity of slope 0.1, and a single output feature in the last layer followed by a sigmoid nonlinearity. We trained the networks in tandem for approximately 10k, 50k and 25k epochs for the 1k, 10k and 100k simulation budgets respectively. There were 10 discriminator updates and 10 generator updates per epoch, with the batch size set to 10% of the simulation budget (also for

100k simulations). All other hyperparameters (learning rate, $\beta_1$, $\beta_2$, etc.) were the same as for the non-optimised architecture.

**Hyperparameters for sequential GATSBI**   The architecture of the generator and discriminator were the same as for amortised GATSBI. We trained the networks for 2 rounds. In the first round, the networks were trained with the same hyperparameters as for amortised GATSBI, on samples from the prior: the only exceptions were the number of samples we held out for the 1k budget (10 instead of 100) and the number of discriminator updates per epoch for the 100k budget (10 instead of 100). Both exceptions were to ensure that there were always 10 discriminator updates per epoch, and speeding up training as much as possible. In the second round, the networks were trained using the energy-based correction with samples from the proposal prior, as well as samples from the prior used in the first round. All other hyperparameters were the same as for round one. The simulation budget was split equally across the two rounds, and the networks were trained anew for each of 10 different observations. The number of epochs was the same for the first and second round: 5k, 10k and 20k for the 1k, 10, and 100k budget respectively. We trained 2 classifiers at the beginning of round two: one to approximate the ratio $\frac{\pi(\theta)}{\tilde{\pi}(\theta)}$ and the other to approximate $\frac{p(x)}{\tilde{p}(x)}$. Both classifiers were 4-layer MLPs with 128 hidden units in each layer, and a ReLU nonlinearity following each layer. The classifiers were trained on samples from the proposal prior and prior, and the proposal marginal likelihood and likelihood respectively, using the MLPClassifier class with default hyperparameters (except for "max_iter"=5000) from scikit-learn (Pedregosa et al., 2011). For the energy-based correction, we used rejection sampling to sample from the corrected distribution $p_t(z)$: for a particular observation $x$, we sampled $z\ p(z) = \mathcal{N}(0,1)$, evaluated the probability of acceptance $p = (\omega(f_\phi(z,x),z))^{-1}/M$, and accepted $z$ if $u < p$ where $u$ is a uniform random variable. To compute the scale factor $M$, we simply took the maximum value of $p(z)\omega(f_\phi(z,x),z))^{-1}$ within each batch. For the inverse importance weights correction, we computed $(\omega(\theta,x))^{-1}$, used it to calculate the loss as in Equation equation 21 for each discriminator and generator update in the second round.

## D.2   SHALLOW WATER MODEL

**Prior**   $\theta \sim \mathcal{N}(\mu\mathbf{1}_{100}, \Sigma),\ \theta \in \mathbb{R}^{100}$
$\mu = 10, \Sigma_{ij} = \sigma\exp(-(i-j)^2/\tau), \sigma = 15, \tau = 100.$

The values for $\boldsymbol{\mu}$ and $\Sigma$ were chosen to ensure that the different depth profile samples produced discernible differences in the corresponding simulated surface waves, particularly in Fourier space. For example, combinations of $\mu$ values $> 25$ (deeper basins), $\sigma$ values $< 10$ and $\tau$ values $> 100$ (smoother depth profiles) resulted in visually indistinguishable surface wave simulations.

**Simulator**

$$\mathbf{x}|\theta = f(\theta) + 0.25\boldsymbol{\epsilon}$$

$$\boldsymbol{\epsilon} = \begin{bmatrix} \epsilon_{1,1} & \cdots & \epsilon_{1,100} \\ \vdots & \ddots & \vdots \\ \epsilon_{200,1} & \cdots & \epsilon_{200,100} \end{bmatrix} \quad \epsilon_{ij} \sim \mathcal{N}(0,1).$$

$f(\theta)$ is obtained by solving the 1D Saint-Venant equations on a 100-element grid, performing a 2D Fourier transform and stacking the the real and imaginary part to form a $2\times100\times100$-dimensional array.

The equations were solved using a semi-implicit solver (Backhaus, 1983) with a weight of 0.5 for each time level, implemented in Fortran (F90). The time-step size $dt$ was set to 300s and the simulation was run for a total of 3600s. The grid spacing $dx$ was 0.01, with dry cells at both boundaries using a depth of $-10$. We used a bottom drag coefficient of 0.001 and gravity=$9.81^m/s^2$. An initial surface disturbance of amplitude 0.2 was injected at $x = 2$, to push the system out of equilibrium.

We chose to perform inference with observations in the Fourier domain for the following reason. Since waves are a naturally periodic delocalised phenomenon, it makes sense to run inference on their Fourier-transformed amplitudes, so that convolutional filters can pick up on localised features. We used the scipy fft2 package (Virtanen et al., 2020).

**GAN architecture**   The generator network was similar to the DCGAN generator (Radford et al., 2015). There were five sequentially stacked blocks of the following form: a 2D convolutional layer, followed by a batch-norm layer and ReLU nonlinearity, except for the last layer, which had only a convolutional layer followed by a `tanh` nonlinearity. The observations input to the generator were of size batch size $\times$ 200 $\times$ 100. The input channels, output channels, kernel size, stride and padding for each of the six convolutional layers were as follows: 1 - (2, 512, 4, 1, 0), 2 - (512, 256, 4, 2, 1), 3 - (256, 128, 4, 2, 1), 4 - (128, 128, 4, 2, 1), 5 - (128, 1, 4, 2, 1). The final block was followed by a fully-connected readout layer that returned a 100-dimensional vector. Additionally, we sampled 25-dimensional noise, where each element of the array was drawn independently from a standard Gaussian. and added it to the output of the `tanh` layer, just before the readout layer.

The discriminator network was similar to the DGCAN discriminator: it contained embedding layers that mirrored the generator network minus the injected noise. The input channels, output channels, kernel size, stride and padding for each of the six convolutional layers in the embedding network were as follows: 1 - (2, 256, 4, 1, 0), 2 - (256, 128, 4, 2, 1), 3 - (128, 64, 4, 2, 1), 4 - (64, 64, 4, 2, 1), 5 - (64, 1, 4, 2, 1). This is followed by 4 fully-connected layers with 256 units each and a leaky ReLU nonlinearity of slope 0.2 after each fully-connected layer. The final fully-connected layer, however, was followed by a sigmoid nonlinearity. The discriminator received both the Fourier-transformed waves and a depth profile alternatively from the generator and the prior as input. The Fourier-transformed waves were passed through the embedding layers, the embedding was concatenated with the input depth profile and then passed through the fully-connected layers.

**Training details**   The two networks were trained in parallel for $\sim$40k epochs, with 100k training samples, of which 100 were held out for testing. We used a batch size of 125, the cross-entropy loss and the Adam optimiser with learning rate$= 0.0001$, $\beta_1 = 0.9$ and $\beta_2 = 0.99$ for both networks. In each epoch, there was 1 discriminator update for every generator update. To ensure stability of training, we used spectral normalisation for the discriminator weights, and clipped the gradient norms for both networks to 0.01, unrolled the discriminator(Metz et al., 2017) with 10 updates i.e., in each epoch, we updated the discriminator 10 times, but reset it to the state after the first update following the generator update.

**NPE and NLE**   We trained NPE and NLE as implemented in the sbi package (Tejero-Cantero et al., 2020) on the shallow water model. We set training hyperparameters as described in Lueckmann et al. (2021), except for the training batch size which we set to 100 for NPE and NLE. The number of hidden units in the density and ratio estimators which we set to 100 (default is 50). For NPE, we included an embedding net to embed the 20k-dimensional observations to the number of hidden units. This embedding net was identical to the one used for the GATSBI discriminators, and it was trained jointly with the corresponding density or ratio estimators. We trained with exactly the same 100k training samples used for GATSBI. MCMC sampling parameters for NLE were set as in Lueckmann et al. (2021).

To calculate **correlation coefficients** for the GATSBI and NPE posteriors, we sampled 1000 depth profiles from the trained networks for each of 1000 different observations from a test set. We then calculated the mean of the 1000 depth profile samples per observation, and computed the correlation coeffiecient of the mean with the corresponding groundtruth depth profile. Thus, we had 1000 different correlation coefficients; we report the mean and the standard deviation for these correlation coefficients.

**Simulation-based calibration** (Talts et al., 2020) offers a way to evaluate simulation-based inference in the absence of ground-truth posteriors. SBC checks whether the approximate posterior $q_\phi(\theta|x)$, when marginalised over multiple observations $x$, converges to the prior $\pi(\theta)$. A posterior that satisfies this condition is well-calibrated, although it is not a sufficient test of the quality of the learned posterior, since a posterior distribution that is equal to the prior would also be well-calibrated. However, when complemented with posterior predictive checks it provides a good test for intractable inference problems.

We performed SBC on the shallow water model. To obtain a test data set $\{\theta_i, x_i\}_{i=1}^N$, we sampled $N = 1000$ parameters $\theta_i$ from the prior and generated corresponding observations $x_i$ from the simulator. For each $x_i$, we then obtained a set of $L = 1000$ posterior samples using the GATSBI generator and calculated the rank of the test parameter $\theta_i$ under the $L$ GATSBI posterior samples

as described in algorithm 1 in Talts et al. (2020), separately for each posterior dimension. For the ranking we used a Gaussian random variable with zero mean and variance 10. We then used bins of $n = 20$ to compute and plot the histogram of the rank statistic. According to SBC, if the marginalised approximate posterior truly matched the prior, the rank statistic for each dimension should be uniformly distributed. Performing SBC can be computationally expensive because the inference has to be repeated for every test data point. In our scenario it was feasible only because GATSBI and NPE perform amortised inference and do not require retraining or MCMC sampling (as in the case of NLE) for every new $x$. We followed the same procedure to do SBC on NPE as for GATSBI.

### D.3 NOISY CAMERA MODEL

**Prior**   The parameters $\theta$ were $28 \times 28$-dimensional images sampled randomly from the 800k images in the EMNIST dataset.

**Simulator**   The simulator takes a clean image as input, and corrupts it by first adding Poisson noise, followed by a convolution with a Gaussian point-spread function: $\mathbf{m} \sim \mathrm{Poisson}(\theta)$
$\mathbf{x}|\mathbf{th} = f * m; \quad f(t) = \exp(-\frac{t^2}{\sigma^2}))$ where $*$ denotes a convolution operation with a series of 1D Gaussian filters given by $f$. We set the width of the Gaussian function $\sigma = 3$.

**GAN architecture**   The generator network was similar to the Pix2Pix generator (Isola et al., 2016): there were 9 blocks stacked sequentially; the first 4 blocks consisted of a 2D convolutional layer, followed by a leaky ReLU nonlinearity with slope 0.2 and a batchnorm layer; the next 4 blocks consisted of transpose a convolutional layer, followed by a leaky ReLU layer of slope 0.2 and a batchnorm layer; the final block had a transposed convolutional layer followed by a sigmoid nonlinearity. The input channels, output channels, kernel size, stride and padding for each of the convolutional or transposed convolutional layers in the 9 blocks were as follows: 1 - (1, 8, 2, 2, 1), 2 - (8, 16, 2, 2, 1), 3 - (16, 32, 2, 2, 1), 4 - (32, 64, 3, 1, 0), 5 - (128, 32, 3, 2, 1), 6 - (64, 16, 2, 2, 1), 7 - (32, 8, 3, 2, 1), 8 - (16, 4, 2, 2, 1), 9 - (4, 1, 1, 1, 0). There were skip-connections from block 1 to block 8, block 2 to block 7, block 3, to block 6 and from block 4 to block 5. 200-dimensional white noise was injected into the fifth block, after convolving it with a 1D convolutional filter and multiplying it with the output of the fourth block.

The discriminator network was again similar to the Pix2Pix discriminator: we concatenated the image from the generator or prior with the noisy image from the simulator, and passed this through 4 blocks consisting of a 2D convolutional layer, a leaky ReLU nonlinearity of slope 0.2 and a batch-norm layer, and finally through a 2D convolutional layer and a sigmoid nonlinearity. The input channels, output channels, kernel size, stride and padding for each of the convolutional were as follows: 1 - (2, 8, 2, 2, 1), 2 - (8, 16, 2, 2, 1), 3 - (16, 32, 2, 2, 1), 4 - (32, 64, 2, 2, 1), 5 - (64, 1, 3, 1, 0).

**Training details**   The generator and discriminator were trained in tandem for 10k epochs, with 800k training samples, of which 100 were held out for testing. We used a batch size of 800, the cross-entropy loss and the Adam optimiser with learning rate= 0.0002, $\beta_1 = 0.5$ and $\beta_2 = 0.99$ for both networks. In each epoch, there was a single discriminator update for every second generator update. To ensure that training was stable, we used spectral normalisation for the discriminator weights, and clipped the gradient norms for both networks to 0.01.

**NPE**   We trained NPE using the implementation in the sbi package (Tejero-Cantero et al., 2020). We set training hyperparameters as described in Lueckmann et al. (2021), except for the training batch size which we set to 1 (in order to ensure that we did not run out of memory while training), and the number of hidden units in the density estimators which we set to 100. $28 \times 28$ dimensional observations were passed directly to the flow, without an embedding net or computing any summary statistics, as was also the case for GATSBI. We trained with exactly the same 800k training samples used for GATSBI.

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
