# OpenReview forum: "GATSBI: Generative Adversarial Training for Simulation-Based Inference"
_ICLR.cc/2022/Conference — ICLR 2022 Poster_

### Official Review · Reviewer_MYcd · 2021-11-02

**Correctness:** 3
**Technical Novelty And Significance:** 3
**Empirical Novelty And Significance:** 3
**Recommendation:** 8
**Confidence:** 4

**Main Review:**

This paper has the potential for broad impact. The number of applications for simulation based inference in science and engineering are far broader than recognized in the ML community.

Much of the prior art, as referenced in the paper, uses ABC, which is really a method of last resort and should be avoided. By presenting a GAN-style alternative to ABC it allows simulation based inference to be applied in much higher dimensional settings than prior methods. The authors show in the paper that the GAN equilibrium distribution is the one that performs exact Bayesian inference through the simulator. It is a clever approach.

Weaknesses:
- The VAE comparison doesn't seem as useful to the reader as more details on how to do actually get training to work. Perhaps there should be a swap of what goes in the appendix.
- The sequential GATSBI setup in eq (7) doesn't make much sense. The reweighting factor requires the computation of the marginal p(x), which is not tractable in general. This setup needs to be clarified.

Questions:
- The paper assumes we do not have access to the likelihood or gradients of the simulator output. In general, the likelihood will be hard to compute for complex simulations. However, with modern autodiff, it is not unreasonable to assume that the gradients may be available even in a complex simulation. Have the authors considered an extension where gradients are available?

Nitpicks:
- Subfigure captions typically go below the figure
- The figures are not BW friendly
- It is typically i.e., not i.e.

**Summary Of The Paper:**

This paper presents a new method to do inference in simulation based systems. It performs "compiled inference" in a sense using a neural network to infer the latent inputs to the simulation. It accomplishes this using a clever GAN-based training objective.

**Summary Of The Review:**

The paper has big potential impact. However, some of the extensions in the paper don't make much sense as presented.

---

> ### Author Response · Authors · 2021-11-12
> **Response to Reviewer MYcd**
>
> Thank you for your detailed review and constructive feedback. We appreciate that you recognize the potential impact of our work, and the advantages of leveraging GATSBI in high-dimensional parameter spaces.
>
> We address below in detail the questions and weaknesses you pointed out:
>
>  > The VAE comparison doesn't seem as useful as getting training to work
>
> We emphasize that elucidating this connection is an important contribution of our paper. It unifies SBI, GANs and adversarial VAEs -- three fields that were previously only held to be conceptually similar -- and provides insights into how to take advantage of those similarities. However, we agree that our algorithms should be usable: to this end, we provide extensive detail on training and network implementation in Appendix D, and plan to release code after the review process. Please also see our response to reviewer DidY
>
> > The sequential GATSBI setup in eq (7) doesn't make much sense. The reweighting factor requires the computation of the marginal p(x)
>
> Thank you for pointing out the lack of clarity in presenting sequential GATSBI.
> The correction factor in equation (7) is $\omega(\theta, x) = \frac{\pi(\theta)}{\tilde{\pi}{\theta}} \frac{\tilde{p}(x)}{p(x)}$. It requires the computation of the following intractable densities: $\tilde{\pi}(\theta)$ the proposal prior; $p(x)$ the marginal likelihood of the observations under the prior $\pi({\theta})$; and $p'(x)$ the marginal likelihood of the observations under the proposal prior. We cannot evaluate these densities, but since we can sample from them, we use the samples to train two ratio density estimators: one for the ratio $\frac{\pi(\theta)}{\tilde{\pi}(\theta)}$ and one for the ratio $\frac{\tilde{p}(x)}{p(x)}$. We then use the ratio density estimators to approximate the correction factor.
> We write about the ratio density estimators in Appendix A4 and D, and will add this clarification to our explanation of sequential GATSBI in section 2.5 of the main paper.
>
> > Have the authors considered an extension [of the simulator] where gradients are available?
>
> We thank you for pointing out a potential and highly intriguing avenue for future work. Indeed, if gradients for the simulator were available, then conditioning the discriminator on the gradients, or using them to guide updates for the generator would potentially improve GATSBI's estimate of the corresponding posterior distribution, and this might be another way in which GATSBI can enrich the toolkit of SBI methods.
>
> > Nitpicks:
>
> Thank you very much indeed for indicating these issues. We have replaced all instances of "i.e." with "i.e.,". We are also currently working on updating the figures to make them more gray-scale friendly, and will post an update when we upload them to the manuscript.
> For the figure with subfigure captions above the figure, could you please indicate which figure you were referring to, so we may fix it?

---

### Official Review · Reviewer_DidY · 2021-11-02

**Correctness:** 4
**Technical Novelty And Significance:** 2
**Empirical Novelty And Significance:** 2
**Recommendation:** 6
**Confidence:** 3

**Main Review:**

I think the novelties of this work are highly related to the importance of the considered simulation-based inference (SBI). Accordingly, it is worth discussing the wide applications of the SBI.

In Section 2.4, it's stated that ``GATSBI, ..., performs an optimization that also finds a minimum of the reverse $D_{KL}$". I think this is an obvious fact, because GATSBI ideally minimizes the JSD whose minimum is identical to that of the reverse $D_{KL}$. I don't understand why using a whole subsection to state that fact?

**Summary Of The Paper:**

This work proposes to exploit GANs to do simulation-based inference (SBI) for black-box simulators. Generally speaking, the proposed GATSBI is identical to do the original GAN in a joint space, with the true distribution $p(x,\theta)=p(\theta)p(x|\theta)$ ($p(x|\theta)$ denotes the simulator) and the fake distribution $q(x,\theta)=p(x)q_{\phi}(\theta|x)$, where $p(x)$ is the marginal of $p(x,\theta)$. The key is that for a black-box simulator, one can query a lot of data pairs $(x,\theta)$.


**Summary Of The Review:**

Technically, the novelties are limited, as both the original GAN and the reverse-KL GAN are well-known in GAN research fields.
However, if the simulation-based inference (SBI) for black-box simulators is quite important in some research fields, then there are certainly contributions from this work.

Empirically, the performance of the proposed GATSBI is only comparable to the baselines.

---

> ### Author Response · Authors · 2021-11-12
> **Response to Reviewer DidY**
>
> Thank you for the thoughtful review. Below, we address in detail your questions about the novelty and significance of our work in the context of SBI, and the relation between GATSBI and GANs.
>
> > I think the novelties of this work are highly related to the importance of the considered simulation-based inference (SBI). Accordingly, it is worth discussing the wide applications of the SBI.
>
> Simulation-based inference is a rapidly expanding field, thanks in part to advances in probabilistic machine learning and to the large applicability of these methods across the fields of science and engineering (Cranmer et al. 2020). In particular, SBI has been applied and led to insights in diverse fields ranging from particle physics to genetics, neuroscience, and epidemiology. However, most previous SBI methods struggle even for low-dimensional parameter spaces (Lueckmann et al. 2021), and for most high-dimensional spaces, benchmarks have not even been attempted. Thus, we believe our contribution is timely and with a potential for real-world applications, where structured problems with large numbers of parameters abound. We note that in the original submission, we have discussed the SBI field and its potential applications both in the introduction and discussion. Furthermore, we have tested GATSBI in a model of high relevance in oceanography. In the revised version, we will expand our discussion about real-world applications.
>
> > performs an optimization that also finds a minimum of the reverse $D_{KL}$[...] I don't understand why using a whole subsection to state that fact?
>
> We have included a subsection on the relation between SBI, GANs and adversarial VAEs, as these fields are highly related but have traditionally not been connected theoretically. By establishing rigorous connections between these subfields of machine learning, we are opening up new avenues for GAN-based approaches to SBI. Furthermore, we would like to explicitly emphasize that GATSBI minimizes a reverse-KL divergence, as (1) it opens up the possibility for future SBI methods to take advantage of the benefits of a reverse-KL divergence objective, and (2) it enabled the natural derivation of a new sequential variant for SBI. Please see also our response to reviewer MYcd.
>
>
> > if the simulation-based inference (SBI) for black-box simulators is quite important in some research fields, then there are certainly contributions from this work.
>
> Thank you for acknowledging the relevance of our work. Simulation-based inference is a rapidly expanding field, see e.g. Cranmer et al 2020, and our discussion of the relevance and applicability of SBI above.
>
>
> > Empirically, the performance of the proposed GATSBI is only comparable to the baselines.
>
> On low-dimensional problems, we do not expect GATSBI to outperform state-of-the-art (SOTA) methods without extensive hyperparameter optimization, as GANs have been shown to struggle in learning 1D distributions (Zaheer et al. 2018). We want point out that, although GATSBI's performance is only on par with SOTA on the high-dimensional shallow water model, it is clearly superior on the camera model, a problem with a high-dimensional and structured parameter space where we expect GATSBI to excel (see also response to reviewer eKE2).

---

### Official Review · Reviewer_cWGX · 2021-11-02

**Correctness:** 3
**Technical Novelty And Significance:** 2
**Empirical Novelty And Significance:** 2
**Recommendation:** 5
**Confidence:** 4

**Main Review:**

I've enjoyed reading the paper: it is written with great attention to detail. The problem is introduced well, the method is clearly motivated and presented in great detail. The highlighted connections to existing methods are nice.

The biggest question mark for me is in novelty. I commend the authors for discussing the connection to LFVI, but this does highlight the fact that the differences between the proposed method and LFVI are arguably minor. Apart from the additional term in the loss (the role of which is unclear), the difference in the used divergence is the main one. As also pointed out by authors in the appendix, the best choice for the divergence is not clear, and also likely problem-specific. It would be interesting to better understand the role of the divergence in such methods, at least empirically.

In addition, work by Adler & Öctem (2018) seems to cover similar ground: a comparable method (but, again, with a difference divergence measure) is proposed, while Adler & Öctem also highlight that their method works with an implicit prior, and evaluate the method on a high-dimensional inverse problem.

Finally, the experimental results are somewhat underwhelming. It's not surprising that we don't win much on the low-dimensional problems, but I wish authors explored more high-dimensional inverse problems, perhaps also more realistic ones.

References:
- Adler & Öctem, Deep Bayesian Inversion, 2018. https://arxiv.org/abs/1811.05910

**Summary Of The Paper:**

Authors propose to use a GAN as an implicit posterior model for performing likelihood-free Bayesian inference, where the generator is trained to sample from the posterior, while the discriminator is trained to distinguish between parameter samples from the prior and the posterior *conditioned on the data sample*. Authors connect the method to prior work in adversarial (variational) inference, and also consider a *sequential* variant of the method, where the posterior approximation is refined iteratively for a particular datapoint. Authors show that the method matches the results of baselines on low-dimensional inference tasks, while also scaling to high-dimensional problems where classic likelihood-free inference methods don't work well.

**Summary Of The Review:**

Overall, while the paper is written excellently, my doubts about novelty and the extent of evaluation are significant enough to put the paper slightly below the bar for acceptance. More extensive evaluation and/or discussion in the context of the prior work would help.

---

> ### Author Response · Authors · 2021-11-12
> **Response to Reviewer cWGX**
>
> Thank you for your thoughtful review and for your appreciation of our work. In addition to the overall response, we address your concerns in more detail below:
>
> > The biggest question mark for me is in novelty.
>
> As we point out in the overall response, we view clarification of connections to existing methods as one of the major strengths of our submission. Moreover, we provide completely novel extensions of GATSBI to sequential variants, initial comparisons against the divergence used in LFVI, and novel experiments comparing GATSBI against state-of-the-art SBI methods. Please see response to eKE2 and DidY as well.
>
> > It would be interesting to better understand the role of the divergence in such methods, at least empirically.
>
> We agree and think that our work already explores this question to some extent: the paper transparently clarifies connections with LFVI and provides some initial comparisons with it. We also discuss the role of the divergence in determining GAN behaviour in Appendix A3. Moreover, we plan an open-source code release after the review period, which will provide an excellent foundation for future endeavors in this direction since a more conclusive answer will require a research project of its own.
>
> > In addition, work by Adler & Öctem (2018) seems to cover similar ground
>
> Thank you very much for pointing us to this relevant work! This work was indeed unknown to us -- it is situated in the medical image reconstruction literature and frames the problem in different terms (as a Bayesian inverse problem), and completely disconnected from both the SBI and the adversarial inference literature (e.g., neither SBI nor adversarial inference papers such as LFVI are cited or empirically compared against in this paper).
>
> Amortised GATSBI is indeed similar to their approach, except for the loss function -- in their case, a Wasserstein metric is used. Their justification for their method rests on stronger assumptions on the GAN networks, i.e., that the discriminator and generator be 1-Lipschitz and K-Lipschitz, respectively. Furthermore, as stated above, a major contribution of the paper is we relate GATSBI to adversarial VAEs and SBI (reverse vs forward KL divergence) via the cross-entropy loss, and also present a sequential variant of this algorithm: neither is as straightforward to derive with the Wasserstein metric.
>
> Furthermore, we think that this just shows the need for connecting and consolidating these disparate ‘islands’ in the literature to provide a basis for future research endeavors, e.g., addressing the question of which divergence to use when. Moreover, the results described in Adler and Öctem (2018) underscore the usefulness of GAN-based SBI approaches for inference in high-dimensional structured parameter spaces.
>
> We will include an extended discussion of this relevant prior work in an updated version of the manuscript and thank you for pointing it out to us.

---

> > ### Comment · Reviewer_cWGX · 2021-11-27
> > **Thank you; still somewhat sceptical**
> >
> > I thank the authors for their engagement with the reviewers and for updating the manuscript. After reading the authors' response and other reviews, I've not changed my mind significantly.
> >
> > The novelty of the GATSBI method by itself is marginal, as pointed out by other reviewers and embraced by the authors. Authors argue, however, that this is offset by the other contributions in the paper.
> >
> > First, the paper aims to rigorously connect the SBI, GANs and adversarial VAEs. Unfortunately, I've not been convinced that there are any special or fundamental connections between these. In the end, one way to treat SBI is as a conditional density estimation task, which we can solve by parametrizing a density, defining a divergence, and minimizing the divergence w.r.t to the parameters of the density model. The density model can be explicit (e.g. a normalizing flow) or implicit (e.g. a GAN, as in the paper). Some divergences can be used in a likelihood-free setting and/or with an implicit density model, others can not. A VAE is simply another instance of a conditional density estimation task, and it so happens that many prior adversarial inference methods have targeted this task instead of SBI. All of this seems well-established in the literature.
> >
> > Second, the paper proposes the sequential variant of GATSBI. I've found this to be the most self-contained contribution: the proposed way to do sequential inference with an implicit posterior density model is neat, and I think the related discussion in the appendix is interesting. However, it is in no way *enabled* by GATSBI, and could have been done in the context of LFVI. Moreover, the paper does not demonstrate empirically when/if doing the sequential adversarial inference is worthwhile.
> >
> > Third, the paper empirically compares GATSBI to other SBI approaches, in particular on two high-dimensional simulators. The goal of these comparisons is not entirely clear to me, however. If the aim is to showcase GATSBI, the results don't provide evidence that we should use it over LFVI. If the aim is to show that GATSBI achieves state-of-the-art results on these tasks, for the Shallow Water Model GATSBI is comparable to the NPE baseline, while for Noisy Camera Model, a standard denoising task with a long history of methods, non-SBI baselines are not included.
> >
> > As a result, I keep my assessment unchanged. I find that the paper tries to do too many things at once, and that doubling down on one or two of the lines of work will make for a stronger contribution. For example, it could be more productive to frame the method as "LFVI with a different divergence", and perform controlled experiments to understand the effect of this. Alternatively, the proposed way to do sequential SBI with an implicit posterior density model is intriguing, and could be made the focus of the paper.

---

> > > ### Author Response · Authors · 2021-11-27
> > > **Thank you**
> > >
> > > We thank the reviewer for their comments and feedback. We maintain that our paper is the first one to rigorously connect literature on SBI with those on GANs and VAEs, and that this is a fundamental contribution of the paper -- while aspects of these relationships have been described in different papers (as we cite and clearly explain),  ours, to the best of our knowledge, is the first one to connect these disparate fields. If we have indeed overlooked a paper, we would be grateful to be pointed to it.
> > >
> > > We thank the reviewer for their acknowledgement of our effort to extensively discuss sequential GATSBI, and its connection to LFVI. It is true that the sequential variant we propose is applicable with either GATSBI or LFVI: we do not claim atany point that it would be exclusive to GATSBI.  Our empirical results are a proof of concept that sequential GATSBI works, but we acknowledge that extensive benchmarking and hyperparameter tuning that would be required to map out when sequential GATSBI would be particularly beneficial.
> > >
> > > As we emphasize in our responses, the two high-dimensional tasks are examples where GATSBI extends the applicability of SBI methods, of which NPE is the current state-of-the-art (rather than just a ‘baseline’). GATSBI’s comparable performance to NPE on the shallow water model is an achievement, since NPE has been tuned by extensive use in the field, while GATSBI was not.  As we state in our manuscript, we set up the noisy camera model as a demonstration for the kinds of problems on which we expect GATSBI to excel, and not to present GATSBI as a replacement for image denoising approaches. GATSBI outperforms NPE here: with a high-dimensional implicit prior, it is not feasible to compare against other SBI methods which require MCMC sampling (e.g., SMC-ABC) or an evaluation of the prior (e.g., NLE).
> > >
> > > As such, these applications were not intended as a comparison between LFVI and GATSBI: indeed, our extensive discussion and empirical comparison of GATSBI, LFVI and Deep Posterior Sampling in Appendix A3 shows that decisions about which method to use can probably only be determined on a case-by-case basis, and would require extensive benchmarking to have more general conclusions.
> > >
> > > It is difficult to change focus and explore divergences and sequential GATSBI in depth, as the reviewer requests, at such a late stage in the review process. We do think that we adequately replied to the request made in the first review ("more extensive evaluation and/or discussion in the context of the prior work would help.") by discussing the work by Adler & Öctem and revising Appendix A.3 (including new results in Figure 6). We maintain that our paper, with its current focus, provides an excellent foundation for further work in SBI.

---

### Official Review · Reviewer_eKE2 · 2021-11-06

**Correctness:** 4
**Technical Novelty And Significance:** 2
**Empirical Novelty And Significance:** 2
**Recommendation:** 6
**Confidence:** 5

**Main Review:**

Strengths:
- This work contributes a new algorithm for SBI for the setup of high-dimensional parameter spaces or implicit priors. This is a significant contribution in itself since this setup is currently out-of-reach for most (if not all) simulation-based inference algorithms.
- The paper is generally very well documented. The position of GATSBI within the context of previous methods for SBI is discussed in detail, much beyond what is usually found in other SBI papers.
- The connection to LFVI is made explicit and discussed in detail in the supplementary materials. Experimental results indicate that both methods perform similarly on the low-dimensional problems.
- The experiments demonstrate the usefulness of GATSBI on two high-dimensional problems. The Noisy Camera model experiment is the most convincing, as it shows its real potential in comparison to other methods.
- The quality of the approximate posteriors is diagnosed with SBC. It reveals that GATBSI provides well-calibrated posteriors.

Weaknesses:
- GATSBI is technically very close to LFVI, proposed initially for likelihood-free in hierarchical implicit models. For this reason, the novelty is limited.
- The experimental validation is limited to two common benchmark problems and two high-dimensional problems. GATSBI does not perform as well as NPE and NLE on the low-dimensional benchmarks.
- GATSBI comes with all the challenges and issues of training GANs.

**Summary Of The Paper:**

This paper introduces GATSBI, an algorithm for simulation-based inference based on adversarial training. This algorithm paves a promising avenue for applying SBI in high-dimensional parameter spaces or with implicit priors -- a setup currently out-of-reach for simulation-based inference algorithms. Experiments deliver a convincing proof-of-concept on a 784-dimensional parameter space, which none of the current SBI algorithms can solve. Experiments on lower-dimensional parameter spaces are less convincing.

**Summary Of The Review:**

In this work, the authors try to address a major obstacle in simulation-based inference. Very few SBI approaches can operate in high-dimensional parameter spaces or with implicit priors, but GATSBI constitutes a convincing proof-of-concept that ought to be investigated further. Experiments are carried out properly, although the experimental validation could have been more thorough. The technical novelty is limited.

Despite some acknowledged weaknesses, I am overall positive about GATSBI. I am willing to recommend it for acceptance.

---

> ### Author Response · Authors · 2021-11-12
> **Response to reviewer eKE2**
>
> Thank you for your careful review. In addition to the general response posted above, we want to address your concerns in more detail below.
>
> > Experiments on lower-dimensional parameter spaces are less convincing.
> > [...] experimental validation could have been more thorough.
>
> Indeed, GATSBI performs less well on lower-dimensional parameter spaces. However, this was expected, as GANs generally struggle for low-dimensional problems (Zaheer et al. 2018). Instead, we expected GATSBI to succeed in regimes where GANs are powerful: in high-dimensional structured spaces. If the reviewers think that it would make the paper more readable, we are happy to move this figure and its description to the supplement.
>
> Regarding a thorough experimental validation: In addition to the validation on a set of public SBI benchmarking problems (Lueckmann et al. 2021), we demonstrated GATSBI on two challenging, intractable high-dimensional problems, the shallow-water model and the camera model. We would like to emphasize that these two problems have not been addressed in the SBI literature before, and that the shallow-water model is a “real-world” model that is of concrete interest to practitioners. On the shallow-water model, we conducted careful experiments with SOTA SBI methods, which in particular, required careful customization of NPE and NRE. We showed that NLE led to poor results, NRE did not converge, and GATSBI performed on par with NPE. Additionally, GATSBI excelled in the camera-model problem, as you acknowledged in your review.
>
>
> > GATSBI is technically very close to LFVI, proposed initially for likelihood-free in hierarchical implicit models.
>
> We agree that LFVI and GATSBI are closely related and discuss this in detail in our submission. However, as we state in our common response to all reviewers, a major contribution of the paper is to connect work in the field of SBI and adversarial inference.  Moreover, we rigorously benchmark our algorithm against other SOTA SBI methods both on low dimensional problems and on high-dimensional problems hitherto unused in SBI, and propose a sequential variant of our algorithm.
>
>
>
> > GATSBI comes with all the challenges and issues of training GANs.
>
> As pointed out in our discussion, the training is indeed more challenging for GATSBI than for other SOTA SBI methods-- a challenge it shares with all other adversarial approaches.  However, we want to emphasize that GATSBI is a first step towards using the power of GANs for SBI. With future research on GATSBI and on GAN training in general (e.g., Sauer et al 2021), we foresee a large potential for GAN-based approaches in SBI, in particular for challenging problems with large numbers of parameters.

---

> > ### Comment · Reviewer_eKE2 · 2021-11-19
> > **Thanks**
> >
> > Thank you for your answer. I remain quite positive about GATSBI. I do not wish to update my evaluation.

---

### Author Response · Authors · 2021-11-12
**Overall response to reviews**

We thank all reviewers for their constructive feedback and insightful comments. We appreciate that all reviewers recognized the potential impact of our submission for simulation-based inference, their positive comments on the presentation of our paper, and that they acknowledged our detailed discussion of related work.

We here briefly address remaining concerns across multiple reviews:

1. Connection to LFVI (Tran et al.):
GATSBI is indeed similar to LFVI in performance and implementation and differs only in the loss function (as pointed out by Reviewers eKE2 and cWGX, and as stated clearly in our submission). However, we emphasize that a primary contribution of our submission is to clarify the connections between SBI methods (see sections 2.1 and 2.3) and adversarial inference approaches (section 2.4). We thereby connected two fields of research that (despite conceptual similarities) had not been rigorously connected yet. Moreover, our paper additionally proposes a sequential variant for GATSBI, empirically compares GATSBI to state-of-the-art SBI approaches, including on two high-dimensional simulators for which SBI had not been applied before.

2. Connection to adversarial variational inference:
Reviewers DiDY and MYcd were concerned about the utility of our theoretical comparisons to adversarial VAEs. We are convinced that this connection is important since the literature concerning SBI, GANs and adversarial VAEs is scattered, and benefits from elucidation of their similarities and differences. Indeed, and as we discussed, some ‘adversarial’ approaches to SBI actually do not correspond to solving (Bayesian) inference problems. By rigorously connecting the two, we open up new avenues for GAN-based approaches to SBI.
Furthermore, the fact that GATSBI implicitly minimizes a reverse-KL divergence bears emphasizing since it opens up interesting research directions for SBI: it allows for SBI methods to take advantage of the benefits of a reverse-KL divergence objective. This objective also allows for the new sequential variant for SBI we present in section 2.5.

3. Experimental results:
Reviewers eKE2 and cWGX expressed concerns about GATSBI's performance on low-dimensional tasks. For this, we refer to Zaheer et al. (2018), who showed that common GAN variants struggle to learn 1D distributions, and thus, we do not expect GATSBI to outperform SOTA methods without extensive hyperparameter optimization.
Reviewer cWGX asked for an exploration of more realistic high-dimensional tasks. We want to emphasize that the shallow water model and noisy camera model were both carefully chosen to demonstrate as examples of the types of structured parameter spaces (common in several fields of science) which are highly challenging for current SBI methods, but for which we expect GATSBI to be useful. Moreover, the shallow water model and its variants are actually ‘real world’ scientific simulation models e.g., frequently used to model tidal motion (Backhaus 1983, Hulscher 1996, Soares-Frazão et al. 2008). We note that these examples are high-dimensional by the standards of SBI, that many SBI methods struggle even on low-dimensional problems (Lueckmann et al., 2021-- many commonly used models there have 10 dimensions or less), and that scaling to high-dimensional problems is one of the outstanding frontiers of SBI (Cranmer et al., 2020).
In summary, we would like to emphasize that the paper’s contributions are _both_ theoretical and empirical: we provide a rigorous explanation for how GANs and SBI are related, we develop new algorithms (GATSBI and sequential GATSBI), and show their relationship to adversarial inference. Furthermore, we introduce two new high-dimensional applications for SBI and demonstrate GATSBI's superior performance on the camera model, and performance on par with SOTA on the shallow water model.

---

### Author Response · Authors · 2021-11-19
**Revised manuscript following reviewer comments**

We thank all reviewers for their valuable feedback.
In addition to our responses to the reviewers' comments, we have now uploaded a revised version of the main and supplementary manuscript with the following changes (highlighted in red in the manuscript):

1. We expand the discussion of SBI applications for GATSBI, the explanation for sequential GATSBI in section 2.5 and update our discussion of related work in section 2.3 to include Adler and Öctem (2018).


2. We have updated Figure 2 of the main manuscript and Figure 6 and 9 of the supplementary material to be more grayscale friendly.


3. We address the connection between GATSBI and Deep Posterior Sampling, presented in Adler and Öctem (2018) in more detail in Appendix A3. We also include results from additional experiments to compare Deep Posterior Sampling with GATSBI and LFVI on the benchmark tasks.
We find that the performance of all three methods is qualitatively similar on these tasks. This adds to our discussion with reviewer cWGX on the value of investigating the role of different divergences in GAN performance. As we state in our responses, an extensive investigation of different divergences is an important avenue for future research, though our analysis here does somewhat address this question.

---

### Author Response · Authors · 2021-11-24
**Responses and updates to manuscript**

We thank the reviewers for taking the time to review our paper and for their valuable feedback. We have responded to the concerns raised in the reviews, and also updated the manuscript with the requested clarifications and additional results. We do hope that these changes address your concerns (reviewer eKE2 has already responded saying that they remain positive about GATSBI and keep their recommendation of ‘accept’). If you have any further questions, please let us know, and  we would be happy to answer them or engage in discussions on the updated material over the next week.

---

### Decision · Program_Chairs · 2022-01-20

**Decision:**

Accept (Poster)

**Comment:**

The paper explores the application of generative adversarial networks as posterior models in simulation-based inference. A new method is proposed, and its connections with related work are studied. The proposed method is empirically evaluated on joint inference of up to 784 parameters.

The reviews are borderline, with one weak reject, two weak accepts, and one strong accept. Overall, the paper is well-written and well-executed. Its main strength is the promising performance of the proposed method in high-dimensional parameter spaces, which are out-of-reach for many existing approaches. The main weakness of the paper is its lack of novelty: the proposed method is only marginally different from already existing ones, while the paper could have explored the differences to a greater extent.

On balance, I'm leaning towards recommending the paper for acceptance. Despite the lack of novelty, the paper is well executed with potential impact in high-dimensional simulation-based inference.